# HiTeA: Hierarchical Temporal Alignment for Training-Free Long-Video Temporal Grounding

**Xinyi Xu**[1], **Hongsong Wang**[2], **Guo-Sen Xie**[3], **Caifeng Shan**[1], **Fang Zhao**[1*]
[1]Nanjing University, [2]Southeast University, [3]Nanjing University of Science and Technology
`xinyi_xu@smail.nju.edu.cn,hongsongwang@seu.edu.cn,gsxiehm@gmail.com,cfshan@nju.edu.cn, fzhao@nju.edu.cn`

## Abstract

Temporal grounding in long, untrimmed videos is critical for real-world video understanding, yet it remains a challenging task owing to complex temporal structures and pervasive visual redundancy. Existing methods rely heavily on supervised training with task-specific annotations, which inherently limits their scalability and adaptability due to the substantial cost of data collection and model retraining. Although a few recent works have explored training-free or zero-shot grounding, they seldom address the unique challenges posed by long videos. In this paper, we propose **HiTeA (Hierarchical Temporal Alignment)**, a novel, training-free framework explicitly designed for long-video temporal grounding. HiTeA introduces a **hierarchical temporal decomposition** mechanism that structures videos into events, scenes, and actions, thereby aligning natural language queries with the most appropriate temporal granularity. Candidate segments are then matched with queries by leveraging **pre-trained vision–language models (VLMs)** to directly compute segment–text similarity, thereby obviating the need for any task-specific training or fine-tuning. Extensive experiments on both short- and long-video benchmarks show that HiTeA not only substantially outperforms all existing training-free methods (e.g., achieving **44.94% R@0.1** on TACoS, representing **an absolute gain of 12.4%**) but also achieves competitive performance against state-of-the-art supervised baselines under stricter metrics. The code is available at `https://github.com/camellia517/HiTea`.

## 1 Introduction

Video Temporal Grounding (VTG) (Gao et al., 2017) aims to locate the start and end timestamps of a video segment that semantically corresponds to a natural language query. This task has broad applications in video surveillance (Tellex & Roy, 2009), sports analytics for event and tactic localization (Kong et al., 2022), and egocentric video understanding (Grauman et al., 2022). However, in contrast to the short, trimmed clips that are predominantly studied in prior work, real-world videos are typically long and untrimmed, characterized by complex temporal structures and significant redundant content. These characteristics render uniform sampling strategies, a mainstay of many existing methods, ineffective at capturing salient moments, thereby constraining their practical scalability.

Current high-performing VTG methods usually rely on fully supervised training with dense temporal labels (Lei et al., 2021; Lin et al., 2023; Huang et al., 2024; Ren et al., 2024). However, the collection of dense temporal annotations is not only costly and subjective but also scales poorly with increasing video length, while the training of large models is computationally intensive. In addition, supervised models are prone to learning dataset-specific biases, leading to limited generalization capability across diverse datasets (Guo et al., 2023). These limitations collectively motivate exploration of training-free approaches under the zero-shot video grounding (ZSVG) setting.

---

*Corresponding author.

Existing ZSVG methods are primarily designed and evaluated for short video clips. Some rely on pseudo-labels to train lightweight models (Nam et al., 2021; Lu et al., 2024; Li et al., 2025), whereas others directly apply pre-trained models without task-specific training (Zheng et al., 2024; Luo et al., 2024), as illustrated in Fig. 1(a). However, such approaches often fail to explicitly model long-range temporal structure, consequently struggling to capture complex temporal dependencies or brief yet critical events in long, untrimmed videos. A promising alternative is to leverage the powerful general-purpose capabilities of large pre-trained video-language models (VLMs) (Bai et al., 2023; Wang et al., 2024b), which excel at video-text alignment. Although VLMs lack the inherent capability to temporally localize events (i.e., *when* they occur), their strength in recognizing semantic content (i.e., *what* happens) provides a crucial prior for grounding, thereby enabling our method to operate in a fully training-free manner.

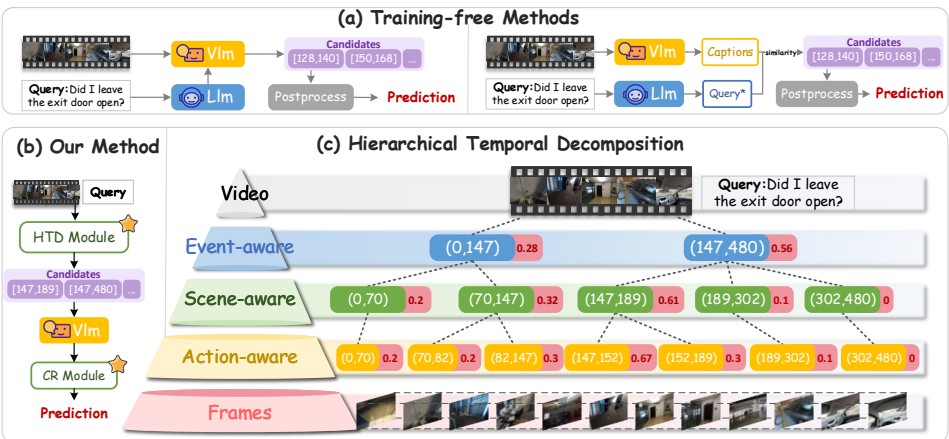

Figure 1: Comparison of different paradigms for zero-shot video temporal grounding. (a) Training-free methods directly use pre-trained models on uniformly sampled frames. (b-c) Our HiTeA framework pipeline and Hierarchical Temporal Decomposition (HTD) module.

These observations motivate a central research question: How can we equip lightweight, off-the-shelf VLMs with explicit temporal structure to achieve accurate grounding in long videos, while entirely foregoing any task-specific training? The core challenge lies in bridging the gap between the VLM's powerful semantic understanding and the need for nuanced temporal reasoning in long-video grounding.

To address this challenge, we propose **HiTeA**, a fully training-free framework inspired by the hierarchical search strategies employed by humans, as shown in Fig. 1(b-c). Instead of processing videos uniformly, HiTeA introduces a Hierarchical Temporal Decomposition (HTD) module, which adaptively partitions a video into multi-scale units, ranging from coarse events to fine-grained actions, using off-the-shelf feature extractors. This constructs a temporal scaffold that emulates human search strategies, making the framework first establish broad contextual understanding before focusing on more informative segments. To enable the efficient application of a frozen VLM to long videos, a pre-filtering step identifies a concise set of candidate segments for subsequent detailed evaluation. Finally, unlike common practices that employ off-the-shelf post-processing like Non-Maximum Suppression (NMS), we design a dedicated Candidate Refinement module that adaptively integrates multi-scale cues from the scored candidates to produce the final, optimal prediction.

Crucially, HiTeA is entirely training-free, eliminating the necessity for any task-specific data collection or model fine-tuning. Experiments show that HiTeA consistently outperforms existing training-free methods on both short- and long-video benchmarks, underscoring that the explicit temporal structure is the key to unlocking the potential of VLMs for precise temporal grounding.

Our main contributions are threefold:

- We propose HiTeA, a **pioneering, fully training-free framework** that effectively addresses temporal grounding in **long videos**.
- HiTeA introduces a **Hierarchical Temporal Decomposition (HTD)** strategy that equips off-the-shelf VLMs with explicit temporal reasoning without any task-specific training.

- Extensive experiments on both short- and long-video benchmarks demonstrate that our approach achieves state-of-the-art performance under the zero-shot setting, with strong generalization.

## 2 RELATED WORK

### 2.1 TRAINING-BASED TEMPORAL GROUNDING IN VIDEOS

The development of video temporal grounding (VTG) has long been dominated by training-based paradigms. Early fully-supervised methods (Zhang et al., 2020; Lin et al., 2023) achieve strong performance but rely on expensive, fine-grained temporal annotations. To reduce labeling costs, subsequent weakly-supervised (Zheng et al., 2022b; Huang et al., 2023) and unsupervised approaches (Wang et al., 2022; Zheng et al., 2023) attempt to learn from weaker signals. Despite alleviating annotation burdens, they still require in-domain training, making them vulnerable to dataset bias and imposing heavy computational costs, particularly when scaling to long-form videos.

Efficiency-oriented designs such as hierarchical or coarse-to-fine architectures (Hou et al., 2022; Pan et al., 2023) have been proposed to mitigate computation, but they remain tied to task-specific training pipelines, limiting their generalizability.

### 2.2 ZERO-SHOT TEMPORAL GROUNDING IN VIDEOS

To overcome annotation and retraining requirements, recent work has explored zero-shot temporal grounding. Existing approaches can be grouped into two streams.

**Pseudo-supervised zero-shot methods** generate automatic pseudo-labels to train grounding models. For example, PSVL (Lu et al., 2024) synthesizes supervisory signals from text corpora and object detectors, attaining accuracy comparable to supervised baselines. However, because they still involve model training, these methods inherit the computational overhead and biases of training-based paradigms.

**Training-free methods** eliminate training altogether, instead leveraging frozen pre-trained models with clever inference strategies, such as query decomposition (Zheng et al., 2024) or boundary-aware proposals (Luo et al., 2024). While computationally lighter, they often lack explicit temporal modeling. Most rely on uniform sampling, which struggles to capture long-range dependencies and can easily miss short but salient events. This leads to inefficiency and performance degradation on complex, long-duration videos.

### 2.3 VIDEO UNDERSTANDING WITH VLMS

The rapid progress of large vision–language models (VLMs) (Bai et al., 2023; Wang et al., 2024b) has further expanded video–text research. VLMs excel at high-level semantic understanding, powering applications like video captioning and question answering. However, they are notably weak at fine-grained temporal localization: VLMs can often tell *what* happens but struggle to determine *when*. To address this gap, recent work augments VLMs with temporal awareness via positional encodings (Guo et al., 2025), timestamp markers (Meinardus et al., 2024b; Chen et al., 2024), or temporal embeddings (Zeng et al., 2024; Ren et al., 2024). These strategies improve short-clip grounding but scale poorly to long videos due to limited context length and rising computation.

## 3 METHODS

### 3.1 OVERVIEW

We propose HiTeA, a training-free framework for temporal grounding in long videos that operates through hierarchical temporal decomposition and query-conditioned matching. As illustrated in Fig. 2, our approach begins with the extraction of multi-scale visual features using pre-trained models such as ViT (Dosovitskiy et al., 2020), DINO (Caron et al., 2021), and RAFT (Teed & Deng, 2020) to capture complementary cues at different levels of abstraction. These features are fed

into a **Hierarchical Temporal Decomposition (HTD)** module to construct a set of candidate segments spanning various temporal granularities. The resulting candidates are subsequently filtered and scored against the text query using a frozen, pre-trained vision–language model (VLM). Finally, a **Candidate Refinement** module merges and ranks these candidates to produce the final temporal predictions, preserving the fully training-free nature of the entire framework.

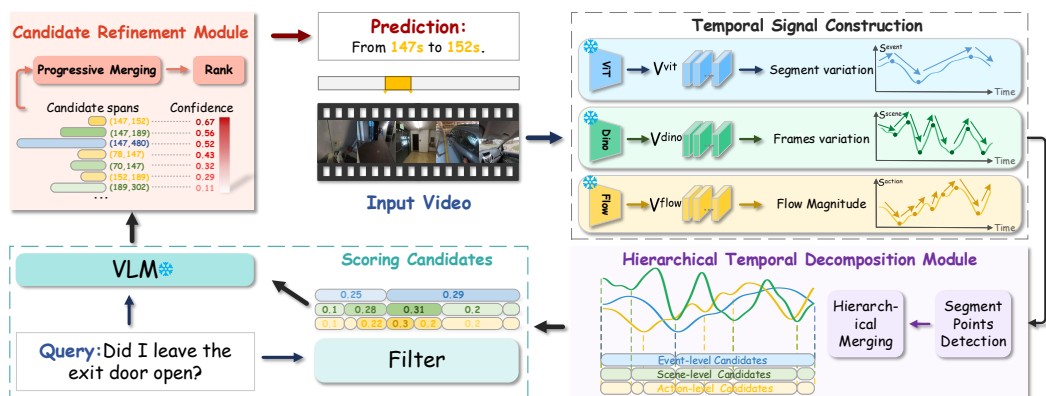

Figure 2: Overview of the HiTeA framework. The Hierarchical Temporal Decomposition(HTD) module decomposes the input video into multi-scale temporal units. Candidate segments are filtered and scored by a pre-trained VLM in a query-conditioned manner. Final moments are generated through Candidate Refinement module without any training.

## 3.2 TEMPORAL SIGNAL CONSTRUCTION

### 3.2.1 FEATURE EXTRACTION.

Given an untrimmed video $V = \{f_1, \ldots, f_T\}$, which consists of $T$ frames, we extract multi-level features from various encoders: ViT for semantic context, DINO for structural transitions, and RAFT for motion dynamics. For frame $f_t$, the extracted features are:

$$\mathbf{v}_t^{\text{vit}} = \phi_{\text{ViT}}(f_t), \quad \mathbf{v}_t^{\text{dino}} = \phi_{\text{DINO}}(f_t), \quad \mathbf{v}_t^{\text{flow}} = \phi_{\text{RAFT}}(f_t, f_{t+1}), \tag{1}$$

where $\phi_{\text{ViT}}$, $\phi_{\text{DINO}}$, and $\phi_{\text{RAFT}}$ are frozen, pre-trained encoders, and $\mathbf{v}_t^{\text{vit}}$, $\mathbf{v}_t^{\text{dino}}$, $\mathbf{v}_t^{\text{flow}}$ are their corresponding event-aware, scene-aware, and action-aware features at time $t$.

### 3.2.2 SIMILARITY CURVES.

To detect potential segment boundaries from extracted multi-level features, we construct three complementary temporal signals, each tailored to capture transitions at a specific temporal granularity.

**Event-level Similarity** ($s^{\text{event}}$): Aimed at mitigating the potential instability of frame-wise ViT features while capturing long-range event transitions, this signal is computed as the cosine similarity between the current frame's feature $\mathbf{v}_t^{\text{vit}}$ and the average feature representation of the most recent video segment. Let $\mathcal{S}_{t-1}$ be the set of frames from the start of the current segment to frame $t-1$.

**Scene-level Similarity** ($s^{\text{scene}}$): This signal is designed to extract the shot-level structure of the video and is computed as the cosine similarity between consecutive frames' DINO features.

**Action-level Similarity** ($s^{\text{action}}$): Since the optical flow feature $\mathbf{v}_t^{\text{flow}}$ inherently represents the motion magnitude between frames $f_t$ and $f_{t+1}$, we derive a motion-aware similarity measure by computing the negative L2 norm of the flow feature. This formulation encourages lower similarity values during high-motion periods, which often correspond to action boundaries.

We define three complementary similarity metrics to capture event, scene, and action-level cues:

$$s_t^{\text{event}} = \frac{\mathbf{v}_t^{\text{vit}} \cdot \bar{\mathbf{v}}_{t-1}^{\text{vit}}}{\|\mathbf{v}_t^{\text{vit}}\|\|\bar{\mathbf{v}}_{t-1}^{\text{vit}}\|}, \quad s_t^{\text{scene}} = \frac{\mathbf{v}_t^{\text{dino}} \cdot \mathbf{v}_{t-1}^{\text{dino}}}{\|\mathbf{v}_t^{\text{dino}}\|\|\mathbf{v}_{t-1}^{\text{dino}}\|}, \quad s_t^{\text{action}} = -\|\mathbf{v}_t^{\text{flow}}\|_2, \tag{2}$$

where $\bar{\mathbf{v}}_{t-1}^{\text{vit}} = \frac{1}{|\mathcal{S}_{t-1}|} \sum_{j \in \mathcal{S}_{t-1}} \mathbf{v}_j^{\text{vit}}$ is the averaged ViT feature over the current segment $\mathcal{S}_{t-1}$. Each signal is smoothed with a Gaussian kernel to suppress noises.

### 3.3 Hierarchical Temporal Decomposition Module

For each similarity curve, candidate boundaries are extracted using granularity-specific strategies: at the event level, boundaries are identified as local minima below a threshold $\tau_k$, whereas at the scene and action levels, the PELT (Killick et al., 2012) change point detection algorithm is employed to capture nonlinear distributional shifts and complex temporal patterns. These candidate points indicate potential temporal transitions.

To enforce the hierarchical containment relation (event $\supset$ scene $\supset$ action), we introduce a merging function $\mathcal{M}(\cdot)$ that integrates higher-level boundaries into lower-level ones. As illustrated in Fig. 3(a), for each higher-level boundary (blue), the nearest lower-level point (red) is replaced if it lies within a temporal tolerance $\alpha$. Otherwise, the higher-level point is inserted. The resulting merged boundaries are highlighted in green. This hierarchical merging process operates sequentially through function $\mathcal{M}(\cdot)$, first combining event-level boundaries into scene-level, then integrating the resulting scene-level boundaries into action-level segments, ultimately yielding the final multi-granularity set $\mathcal{P}^{\text{final}}$.

This hierarchical design guarantees that fine-grained action segments remain aligned with broader structures. However, strict structural enforcement primarily benefits long videos with complex nesting. For short videos lacking deep hierarchies, such constraints may filter valid candidates. Thus, we adopt an **adaptive strategy**: applying $\mathcal{M}(\cdot)$ to long videos for coherence, while bypassing it for short videos to preserve boundary diversity (See in Appendix A.4.2).

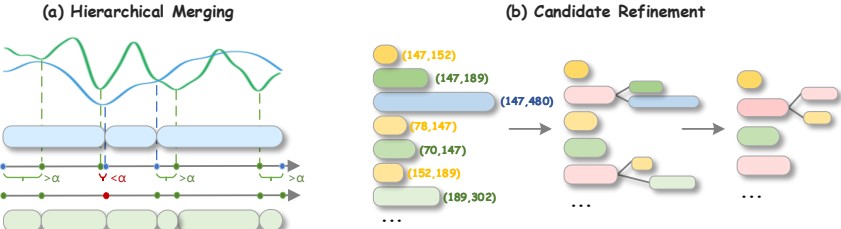

Figure 3: Key stages of our approach. (a) Hierarchical merging to form candidate segments. (b) Candidate refinement to achieve accurate localization.

### 3.4 Scoring Candidates

Directly applying VLM to all candidate segments is computationally expensive, particularly for long videos with dense proposals. To strike a balance between accuracy and efficiency, we adopt a two-stage scoring strategy that progressively prioritizes high-quality candidates.

**VideoCLIP-based Filtering.** We first compute a coarse similarity score $s_{\text{clip}}$ between each candidate segment $(t_s, t_e)$ and the query $Q$ using a VideoCLIP (Xu et al., 2021; Wang et al., 2024a), which is designed for contrastive learning between video clips and text. To reduce fragmentation and enhance semantic continuity, we merge adjacent segments with similar scores. Specifically, two neighboring segments $i$ and $j$ are merged if $|s_{\text{clip}}^i - s_{\text{clip}}^j| < \beta$, where $\beta$ is a threshold. From the merged set, we retain the top-$k$ segments per hierarchy based on $s_{\text{clip}}$. This step significantly prunes the candidate pool while preserving segments with a high likelihood of matching the query.

**VLM-based Scoring.** Then, the retained candidates are evaluated using a frozen, off-the-shelf VLM to obtain fine-grained similarity scores $s_{\text{vlm}} \in [0, 1]$. Although absolute scores may vary with prompt design, the relative rankings across candidates remain stable, enabling reliable candidate selection. This stage leverages the strong semantic alignment of VLMs in a fully training-free manner, combining the efficiency of VideoCLIP-based pre-filtering.

### 3.5 Candidate Refinement Module

We observe that the reliability of VLM-based scoring is sensitive to both segment duration and prompt formulation. Extremely long or short segments tend to yield unreliable scores due to frame sampling limitations or uneven content density, while prompt semantics can introduce systematic

biases. To mitigate these issues and explicitly adapt to the HTD-generated candidates, we propose a refinement module that, distinct from conventional post-processing, integrates cross-modal similarity with intra-video consistency, followed by progressive merging and ranking.

**Score Fusion.** Each candidate segment $c_i$ is assigned a unified relevance score by combining the VLM-based similarity and the VideoCLIP-based temporal consistency. While the VLM provides superior semantic reasoning, its output confidence scores often exhibit discretization, resulting in tied rankings among candidate segments. To mitigate this, we leverage the continuous nature of $s_{\text{clip}}$ to provide fine-grained discrimination. This fusion strategy ensures that the VLM's semantic alignment remains dominant, while $s_{\text{clip}}$ serves as a critical tie-breaker, resolving scoring ambiguities without compromising the primary semantic order.

Before fusion, the $s_{\text{clip}}$ is normalized to the range [0, 1] to ensure scale compatibility. The final score is computed as:

$$s_{\text{final}} = \lambda \cdot s_{\text{vlm}} + (1 - \lambda) \cdot s_{\text{clip}} , \qquad (3)$$

where $\lambda$ balances the two terms. This fusion mitigates the impact of outlier scores and reinforces segments that are both semantically relevant and structurally coherent.

**Progressive Merging.** To exploit complementary information across hierarchical levels (e.g., an action within a relevant scene), we introduce a query-aware, progressive merging strategy that enables cross-level interaction, in Fig. 3(b). Segments at different granularities—actions, scenes, and events—are considered jointly during merging, guided by their semantic relevance to the query. Specifically, two candidates from the same or adjacent levels are merged if they are temporally close and have high query-conditioned similarity. The merging decision accounts for both boundary proximity and consistency of their VLM scores, ensuring semantic coherence in the merged segment. This process is applied iteratively, allowing, for example, a high-scoring action segment to merge with an overlapping scene segment to form a richer candidate that integrates multi-level cues. The merged segment inherits a refined confidence score computed from its constituents, reinforcing segments that benefit from cross-hierarchical fusion. Further details are in Appendix A.2.2.

**Ranking and Output.** The refined set of temporal segments $\mathcal{S}_{\text{final}} = \{c_i\}$ is ranked according to their updated scores $s_{\text{final}}^{(i)}$, and the top-ranked segment is selected as the final prediction:

$$(\hat{t}_s, \hat{t}_e) = \arg \max_{c_i \in \mathcal{S}_{\text{final}}} s_{\text{final}}^{(i)} . \qquad (4)$$

This refinement stage effectively complements HTD module by leveraging cross-level information and the scores from the previous stage to generate temporal predictions.

## 4 EXPERIMENTS

### 4.1 EXPERIMENTAL SETTINGS

**Datasets.** We evaluate HiTeA on two categories of benchmarks. For **short-video** temporal grounding, we use **Charades-STA** (Sigurdsson et al., 2016), originally introduced for videoQA, comprising 1.3K test videos with 3.7K queries of household activities, **QVHighlights** (Lei et al., 2021), containing over 1.5K validation and 1.5K test videos with diverse topics that require models to detect all relevant temporal spans and their saliency scores, and **ActivityNet-Captions** (Krishna et al., 2017), repurposed from dense video captioning, containing 4.9K open-domain videos with 17.0K annotated descriptions. For **long-video** temporal grounding, we use **Ego4D-NLQ** (Grauman et al., 2022) with 415 egocentric validation videos, and **TACoS** (Regneri et al., 2013), a compact, densely-annotated cooking dataset with 25 test videos.

**Evaluation Metrics.** Following established practices in temporal grounding, we report Recall@1 at multiple IoU thresholds and the mean Intersection-over-Union (mIoU). The evaluation thresholds are set to 0.3, 0.5, 0.7 for short videos and 0.1, 0.3, 0.5 for long videos, reflecting the higher complexity of long videos and the difficulty of accurately retrieving very short segments with current zero-shot methods. The mIoU metric complements these by providing an overall accuracy measure, calculated by averaging the IoU across all predictions.

Table 1: Performance comparison on **long-video** temporal grounding benchmarks. 'Sup' denotes the supervision type: FS (Fully-Supervised), ZS (Zero-Shot). [†] Method requires training.

| Method | Sup | Ego4D-NLQ | | | | TACoS | | | |
|---|---|---|---|---|---|---|---|---|---|
| | | 0.1 | 0.3 | 0.5 | mIoU | 0.1 | 0.3 | 0.5 | mIoU |
| 2D-TAN (Zhang et al., 2020) | FS | - | 4.33 | 1.83 | 3.39 | 47.59 | 37.29 | 25.32 | - |
| Moment-DETR (Lei et al., 2021) | FS | - | 4.34 | 1.81 | 3.53 | - | 24.67 | 11.97 | 25.49 |
| UniVTG (Lin et al., 2023) | FS | - | 7.28 | 3.95 | 4.91 | - | 51.44 | 34.97 | 33.60 |
| UniVTG (Lin et al., 2023) [†] | ZS | - | 6.48 | 3.48 | 4.63 | - | 5.17 | 1.27 | 4.40 |
| Mr.BLIP (Meinardus et al., 2024a) [†] | ZS | - | 6.49 | 3.20 | 5.37 | - | 24.59 | 14.32 | 17.94 |
| TimeSuite (Zeng et al., 2024) [†] | ZS | - | 0.88 | 0.43 | 0.94 | - | 6.75 | 2.50 | 5.71 |
| UniTime-Zero (Li et al., 2025) [†] | ZS | - | 14.67 | 7.38 | 10.18 | - | 50.06 | 31.54 | 33.38 |
| (Luo et al., 2024) | ZS | - | - | - | - | 27.49 | 11.20 | 5.57 | - |
| **HiTeA (Qwen2.5-VL)** | ZS | **20.12** | **10.39** | **6.04** | **8.12** | **44.94** | **29.08** | **16.10** | **19.79** |

Table 2: Performance comparison on **short-video** temporal grounding benchmarks. Supervision (Sup): FS, WS (Weakly-Supervised), US (Unsupervised), ZS. [†] Method requires training

| Method | Sup | Charades-STA | | | | ANet-Captions | | | |
|---|---|---|---|---|---|---|---|---|---|
| | | 0.3 | 0.5 | 0.7 | mIoU | 0.3 | 0.5 | 0.7 | mIoU |
| 2D-TAN (Zhang et al., 2020) | FS | 57.3 | 45.8 | 27.9 | 41.0 | 60.4 | 43.4 | 25.0 | 42.5 |
| Moment-DETR (Lei et al., 2021) | FS | 62.1 | 48.2 | 25.3 | 42.3 | 52.6 | 32.5 | 15.3 | 37.8 |
| VTimeLLM (Huang et al., 2024) | FS | 55.3 | 34.3 | 14.7 | 34.6 | 44.8 | 29.5 | 14.2 | 31.4 |
| CPL (Zheng et al., 2022b) | WS | 66.40 | 49.24 | 22.39 | - | 55.73 | 31.37 | - | - |
| CNM (Zheng et al., 2022a) | WS | 60.39 | 35.43 | 15.45 | - | 55.68 | 33.33 | - | - |
| (Huang et al., 2023) | WS | 69.16 | 52.18 | 23.94 | 45.20 | 58.07 | 36.91 | - | 41.02 |
| PSVL (Nam et al., 2021) | US | 46.47 | 31.29 | 14.17 | 31.24 | 44.74 | 30.08 | 14.74 | 29.62 |
| PZVMR (Wang et al., 2022) | US | 46.83 | 33.21 | 18.51 | 32.62 | 45.73 | 31.26 | 17.84 | 30.35 |
| DSCNet (Liu et al., 2022) | US | 44.15 | 28.73 | 14.67 | - | 47.29 | 28.1 | - | - |
| SPL (Zheng et al., 2023) | US | 60.73 | 40.70 | 19.62 | 40.47 | 50.24 | 27.24 | 15.03 | 35.44 |
| UniVTG (Lin et al., 2023) [†] | ZS | 44.09 | 25.22 | 10.03 | 27.12 | - | 11.10 | 4.06 | 16.86 |
| VideoChatGPT-7B (Maaz et al., 2023) [†] | ZS | 20.0 | 7.7 | 1.7 | 13.7 | 26.4 | 13.6 | 6.1 | 18.9 |
| VTG-GPT (Xu et al., 2024) [†] | ZS | 59.48 | 43.68 | 25.94 | 39.81 | 47.13 | 28.25 | 12.84 | 30.49 |
| (Lu et al., 2024) [†] | ZS | 47.74 | 34.62 | 20.16 | 32.97 | 49.26 | 31.45 | 15.27 | 33.25 |
| UniTime-Zero (Li et al., 2025) [†] | ZS | - | 59.09 | 31.88 | 52.19 | - | 22.77 | 14.14 | 27.31 |
| (Luo et al., 2024) | ZS | 56.77 | 42.93 | 20.13 | 37.92 | 48.28 | 27.90 | 11.57 | 32.37 |
| TFVTG (Zheng et al., 2024) | ZS | 67.04 | **49.97** | 24.32 | 44.51 | 49.34 | 27.02 | 13.39 | 34.10 |
| (Xu et al., 2025) | ZS | 58.2 | 38.4 | 21.6 | 36.5 | 48.1 | 31.1 | 14.9 | 30.8 |
| **HiTeA (Qwen2.5-VL)** | ZS | **69.62** | 48.52 | **25.59** | **46.29** | **54.46** | **31.1** | **16.57** | **37.93** |

**Implementation Details.** Our framework is implemented in PyTorch. We use Qwen2.5-VL-7B (Bai et al., 2023) as the base VLM, and VideoCLIP-XL (Wang et al., 2024a) for pre-filtering. For feature extraction, we employ ViT-B/32 for event-level features, DINO-v2 for scene-level features, and RAFT-Large for action-level features. The frame sampling rate is set to 5 FPS for Charades-STA and 1 FPS for all other datasets. Most hyperparameters remain unchanged across all datasets.

**Adaptive Configuration.** We adjust the Hierarchical Merging (HM) module based on video duration. HM is **enabled** for long-video datasets to enforce structural consistency, and **disabled** for short-video datasets to maximize candidate diversity from different feature levels. Additional implementation details are provided in Appendix A.3, while supplementary experiments, including analyses on model selections and hierarchical merging effects, are in Appendix A.4.

## 4.2 COMPARISON WITH THE STATE-OF-THE-ARTS

HiTeA establishes a new state-of-the-art for zero-shot temporal grounding, consistently outperforming prior methods on both long- and short-video benchmarks (Tables 1 and 2).

Table 3: Performance comparison on **QVHighlights** temporal grounding benchmarks. Supervision (Sup): FS, WS (Weakly-Supervised), US (Unsupervised), ZS. [†] Method requires training

| Method | Sup | Test | | | | Val | | | |
|---|---|---|---|---|---|---|---|---|---|
| | | R1@0.5 | R1@0.7 | mAP@0.5 | mAP@avg | R1@0.5 | R1@0.7 | mAP@0.5 | mAP@avg |
| Moment-DETR (Lei et al., 2021) | FS | 52.9 | 33.0 | 54.8 | 30.7 | 54.2 | 33.4 | 55.4 | 31.1 |
| UniVTG (Lin et al., 2023) | FS | 58.9 | 40.9 | - | - | - | - | - | - |
| VTimeLLM (Huang et al., 2024) | FS | 47.2 | 29.3 | 47.3 | 27.4 | 48.8 | 29.5 | 49.3 | 26.8 |
| Mr.BLIP (Meinardus et al., 2024a) | FS | 74.8 | 60.5 | 76.1 | 63.4 | - | - | - | - |
| CNM (Zheng et al., 2022a) | WS | 14.1 | 4.0 | 11.8 | - | - | - | - | - |
| CPL (Zheng et al., 2022b) | WS | 30.8 | 10.8 | 22.8 | - | - | - | - | - |
| CPI (Kong et al., 2023) | WS | 32.3 | 11.8 | 23.7 | - | - | - | - | - |
| PZVMR (Wang et al., 2022) | US | 14.2 | 4.9 | 15.7 | 4.6 | 12.6 | 5.1 | 16.2 | 5.3 |
| DSCNet (Liu et al., 2022) | US | - | - | - | - | 12.3 | 3.5 | 10.4 | 2.7 |
| TimeSuite (Zeng et al., 2024) [†] | ZS | 12.3 | 9.2 | - | - | - | - | - | - |
| VTimeLLM(Huang et al., 2024) [†] | ZS | 26.1 | 11.2 | - | - | - | - | - | - |
| UniTime-Zero (Li et al., 2025) [†] | ZS | 41.0 | 31.5 | - | - | - | - | - | - |
| (Diwan et al., 2023) | ZS | - | - | - | - | 48.3 | 31.0 | 47.3 | 28.0 |
| Moment-GPT (Xu et al., 2025) | ZS | 58.3 | 37.7 | 55.1 | 35.0 | 58.9 | 38.6 | 55.7 | 35.9 |
| **HiTeA (Qwen2.5-VL)** | ZS | **62.3** | **42.2** | **60.7** | **37.0** | **64.1** | **43.2** | **60.2** | **37.3** |

**Performance on Long Videos.** HiTeA sets a new state-of-the-art for zero-shot temporal grounding on long videos. A groundbreaking achievement is made on the Ego4D-NLQ benchmark: it achieves an mIoU of 8.12%, surpassing all fully-supervised baselines. To our knowledge, this is the first time a fully training-free method has been successfully applied to this dataset, and it delivers superior performance. On TACoS, HiTeA also dominates, achieving 44.94% at R@0.1—nearly doubling the best prior training-free result. These results demonstrate that our hierarchical temporal decomposition (HTD) strategy effectively captures the complex temporal structures of long videos, proving that explicit hierarchical reasoning can outperform data-intensive supervised training.

**Generalization to Short Videos.** Notably, although HiTeA is specifically designed for long videos, it also generalizes remarkably well to short-video settings. On Charades-STA, it achieves 69.62% at R@0.1 and 25.59% at R@0.5, surpassing all existing zero-shot approaches and even matching several weakly- and fully-supervised methods. On ActivityNet-Captions, HiTeA also demonstrates clear superiority, outperforming the strongest zero-shot baseline. Furthermore, in QVHighlights, HiTeA performs significantly well across both test and validation splits. On the Test split, HiTeA achieves 60.7% at mAP@0.5 and 37.0% at mAP@avg, which is a strong result, surpassing many of the state-of-the-art zero-shot methods.

The consistent gains across datasets of varying durations and complexity confirm the efficacy of HiTeA's training-free architecture, particularly the synergy between hierarchical candidate generation and query-aware VLM scoring. Our approach effectively bridges the gap between semantic understanding and temporal localization without task-specific training.

## 4.3 ABLATION STUDIES

To evaluate the contribution of each component, we conduct comprehensive ablation studies on the TACoS and Charades-STA datasets, representing long and short video benchmarks, respectively.

### 4.3.1 EFFECTIVENESS OF INDIVIDUAL MODULES

Table 4 reports the ablation results for Hierarchical Temporal Decomposition (HTD) and Candidate Refinement (CR) on TACoS and Charades-STA. The baseline without HTD or CR (first row), which applies uniform segmentation, achieves the lowest performance, demonstrating that naive temporal partitioning fails to capture complex temporal dynamics in both long and short videos. For configurations *without* CR, the model selects the single candidate with the highest fusion score. Conversely,

Table 4: Ablation Study on Hierarchical Temporal Decomposition and Candidate Refinement.

| HTD | | | CR | TACoS | | | | Charades-STA | | | |
|---|---|---|---|---|---|---|---|---|---|---|---|
| Event | Scene | Action | | 0.1 | 0.3 | 0.5 | mIoU | 0.3 | 0.5 | 0.7 | mIoU |
| - | - | - | - | 41.18 | 21.41 | 8.25 | 14.87 | 55.65 | 33.17 | 15.11 | 35.07 |
| ✓ | - | - | - | 39.34 | 21.29 | 10.76 | 15.64 | 63.92 | 40.00 | 20.94 | 41.28 |
| - | ✓ | - | - | 30.21 | 18.78 | 10.21 | 13.01 | 59.89 | 38.82 | 20.75 | 40.35 |
| - | - | ✓ | - | 30.85 | 19.74 | 11.90 | 13.73 | 60.13 | 40.03 | 20.38 | 39.83 |
| - | ✓ | ✓ | - | 34.32 | 22.11 | 13.53 | 15.34 | 60.81 | 41.37 | 22.18 | 40.76 |
| ✓ | - | ✓ | - | 35.52 | 22.25 | 13.07 | 15.61 | 62.82 | 43.04 | 21.69 | 41.40 |
| ✓ | ✓ | - | - | 34.97 | 21.49 | 11.37 | 14.98 | 63.31 | 42.58 | 23.01 | 41.85 |
| ✓ | ✓ | ✓ | - | 38.44 | 24.41 | 14.35 | 16.99 | 62.69 | 43.31 | 22.66 | 41.61 |
| - | - | - | ✓ | 45.03 | 23.21 | 10.32 | 16.67 | 67.63 | 39.46 | 20.00 | 43.93 |
| ✓ | ✓ | ✓ | ✓ | **44.94** | **29.08** | **16.10** | **19.79** | **69.62** | **48.52** | **25.59** | **46.29** |

enabling CR activates the iterative merging process, allowing adjacent segments to be aggregated based on semantic similarity to form the final prediction.

**Hierarchical Temporal Decomposition.** Examining individual HTD layers reveals their complementary strengths. **Event-level** segmentation achieves high recall at loose thresholds (39.34% R@0.1 on TACoS, 63.92% R@0.3 on Charades-STA), reflecting the fact that coarse segments are more likely to overlap with ground-truth intervals. However, its precision decreases at stricter thresholds, indicating limited ability to accurately localize fine-grained boundaries. In contrast, **Action-level** segmentation achieves the highest R@0.5 on TACoS (11.90%), highlighting its utility for precise temporal localization. **Scene-level** segmentation underperforms on both datasets, likely because TACoS and Charades-STA focus on cooking or household activities, where scene transitions are rare and most queries do not correspond to scene changes.

Combining multiple layers consistently improves performance, illustrating the complementarity of different temporal granularities. **Two-level** combinations, such as Scene+Action, consistently outperform any single layer by capturing both coarse coverage and fine-grained details. Extending the decomposition to **three levels** yields further gains on TACoS (mIoU: 16.99%), whereas the improvement on Charades-STA is marginal (mIoU:41.61%). This indicates that multi-level decomposition is most beneficial in long videos, where temporal structures are more complex, while short videos are often simple enough that a single or two-level decomposition suffices.

**Candidate Refinement.** The CR module, designed to adaptively refine the candidates produced by HTD, brings consistent and substantial gains across benchmarks. It boosts mIoU from 16.99% to 19.79% on TACoS and from 41.61% to 46.29% on Charades-STA, with gains observed at all IoU thresholds. Comparing uniform + w/o CR and uniform + w CR, we observe that CR leads to significant gains. The uniform + w/o CR baseline performs worse due to poor segmentation, while uniform + w CR improves mIoU by refining boundaries. The full all w/CR configuration further enhances performance, confirming CR's role in handling ambiguities and providing more accurate segmentation. These results demonstrate that CR effectively filters noisy segments and sharpens temporal boundaries, confirming its role as a crucial complement to HTD by exploiting cross-level interactions and query-aware merging for more robust and precise localization.

### 4.3.2 EFFICIENCY OF VIDEOCLIP-BASED PRE-FILTERING

We quantitatively evaluate the efficiency of our pre-filtering stage, designed to avoid the computational cost of exhaustively scoring all candidate segments with VLMs,shown in Table 5. The reduction ratio correlates with video length and structural complexity. On Ego4D-NLQ, featuring long egocentric videos with dense content, filtering reduces the average number of segments from 146.68 to 9.0 per video, corresponding to a 93.9% reduction. Similarly, TACoS and ActivityNet-Captions see reductions of 76.6% and 79.4%, respectively.

These results confirm that our filtering preserves high-quality candidates while pruning irrelevant ones, reducing the number of VLM score computations from hundreds to single digits per video.

Table 5: Efficiency Improvement of VideoCLIP-based Filtering

| Measure | Charades-STA | ANet-Captions | TACoS | Ego4D-NLQ |
|---|---|---|---|---|
| Segments (w/o Filter) | 9.75 | 22.26 | 38.19 | 146.68 |
| Segments (w/ Filter) | 8.17 | 4.59 | 8.92 | 9.0 |
| **Reduction (%)** | **16.2** | **79.4** | **76.6** | **93.9** |

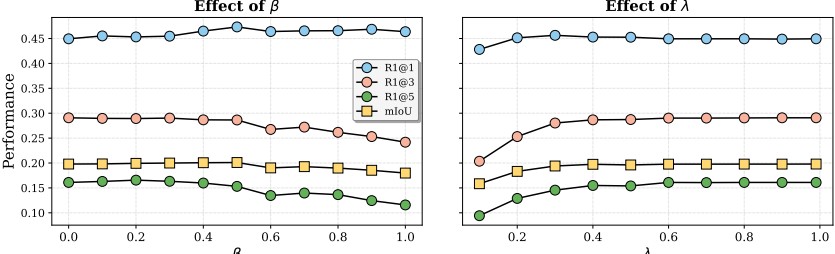

Figure 4: Parameter sensitivity analysis. Left: the effect of $\beta$ on VideoCLIP-based pre-filtering. Right: the effect of $\lambda$ on the final scoring function.

This makes training-free temporal grounding computationally tractable and scalable, particularly for long and complex videos.

### 4.4 PARAMETER SENSITIVITY

We analyze the sensitivity of our method on the TACoS dataset with respect to two key hyperparameters: the merging threshold $\beta$ in the VideoCLIP-based pre-filtering stage and the weighting parameter $\lambda$ in the CR module. Figure 4 reports Recall@$K$ and mIoU in different parameter values.

**For $\beta$ (Figure 4, left)**, performance remains relatively stable in the range $[0.2, 0.6]$, with the best mIoU observed at $\beta = 0.5$. This indicates that the pre-filtering threshold is not highly sensitive, since the hierarchical decomposition already provides diverse proposals. However, when $\beta$ exceeds 0.6, semantically distinct moments tend to be merged, which harms localization accuracy.

**For $\lambda$ (Figure 4, right)**, the highest mIoU is achieved at $\lambda = 0.99$. This validates that the VLM score $s_{\text{vlm}}$ is the dominant factor for accurate localization. However, as the VLM tends to output discrete confidence values, the VideoCLIP score $s_{\text{clip}}$ provides essential complementary cues. By assigning a minimal weight (1%) to $s_{\text{clip}}$, we effectively break ties among candidates with equal VLM scores without disrupting the high-quality semantic alignment provided by the VLM.

### 4.5 INFERENCE EFFICIENCY

To evaluate scalability, we benchmarked HiTeA against representative baselines on both long-video and short-video datasets. In our analysis, HiTeA achieves a stable online inference latency ($\sim$7-8s) that is largely insensitive to video length. This significantly outperforms dense scanning or proposal-based baselines, which often suffer from linear or quadratic complexity scaling on long inputs. The detailed runtime analysis and full comparison table are provided in Appendix A.6.

## 5 CONCLUSION

We have presented HiTeA, a fully training-free framework for temporal grounding in long videos, designed to tackle the challenges of complex temporal structures and high computational costs. Central to our approach is the Hierarchical Temporal Decomposition (HTD) strategy, which structures videos into a coarse-to-fine hierarchy. This allows off-the-shelf VLMs to localize queries with high precision, completely without supervision. Extensive experiments validate our approach, showing state-of-the-art zero-shot performance. HiTeA's strong results on long-video benchmarks highlight its scalability and robustness, paving the way for practical, efficient video understanding systems.

## REPRODUCIBILITY STATEMENT

To ensure reproducibility, we have taken several measures. A complete description of the proposed framework, and further implementation details, including feature extraction, hyperparameter settings, and dataset-specific preprocessing, are documented in Appendix. The full source code is available at the link: `https://github.com/camellia517/HiTea`. While pre-trained models and raw datasets must be obtained from their official sources due to size constraints, we provide examples of processed data formats and will release all intermediate files generated during preprocessing to make replication straightforward. In addition, all pre-trained models used in our work are publicly available, with their versions and usage explicitly specified in the appendix. These resources are intended to facilitate faithful reproduction of our results and foster future research in zero-shot temporal grounding.

## ACKNOWLEDGMENTS

This work was supported by the National Natural Science Foundation of China (No. 62476124, 62276134, 62172090), Fundamental and Interdisciplinary Disciplines Breakthrough Plan of the Ministry of Education of China (JYB2025XDXM902), Natural Science Foundation of Jiangsu Province (No. BK20242015, BK20230833), and the Gusu Innovation and Entrepreneur Leading Talents (No. ZXL2025322).

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

# A APPENDIX

## A.1 QUALITATIVE RESULTS.

### A.1.1 LONG-VIDEO EXAMPLE

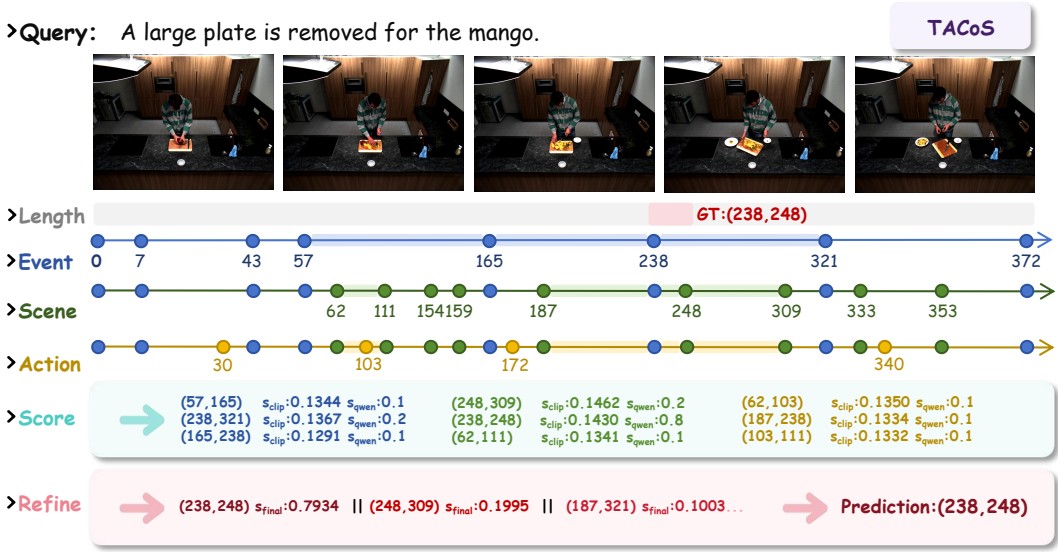

Figure 5: A qualitative example of the proposed hierarchical temporal decomposition and matching process for the query.

Figure 5 illustrates the hierarchical reasoning of our framework for the query *"A large plate is removed for the mango."* Multi-scale candidate boundaries are first generated by the HTD module—event-level (blue), scene-level (green), and action-level (yellow) splits. Irrelevant candidates are filtered by VideoCLIP, and the remaining segments are scored by a frozen VLM (Qwen2.5-VL), where $s_{\text{clip}}$ provides coarse semantic alignment and $s_{\text{qwen}}$ reflects more precise query-conditioned relevance. The CR module then integrates and re-ranks these candidates to produce the final prediction.

In this example, the short but critical interval *[238, 248]*—corresponding to the instantaneous plate removal—receives the highest confidence score ($s = 0.79$) after refinement, whereas longer and more ambiguous segments (e.g., *[248, 309]*) are assigned lower scores. This demonstrates that our framework effectively localizes fleeting yet semantically important events in long videos by combining hierarchical temporal reasoning with coarse-to-fine semantic matching from a general-purpose VLM, all without any task-specific training.

### A.1.2 SHORT-VIDEO EXAMPLES

Figure 6 presents two qualitative examples from short-video datasets (Charades-STA). Since short videos typically lack significant scene-level changes, we selected two illustrative clips to showcase different decomposition strategies: single-level action decomposition (top) and two-level action+event decomposition (bottom).

The top example demonstrates that action-level decomposition alone can successfully capture fine-grained temporal transitions. Despite only three action segments and relatively low VideoCLIP scores, the relevant candidates are retained, and the frozen Qwen VLM accurately matches the textual query to the corresponding video segments.

In the bottom example, combining event-level and action-level decompositions captures semantic variations more comprehensively, resulting in precise segment boundaries. Along with Figure 5, these examples highlight that our framework can flexibly adapt to videos of different lengths, demonstrating robust localization capability and strong generalizability across both short and long videos.

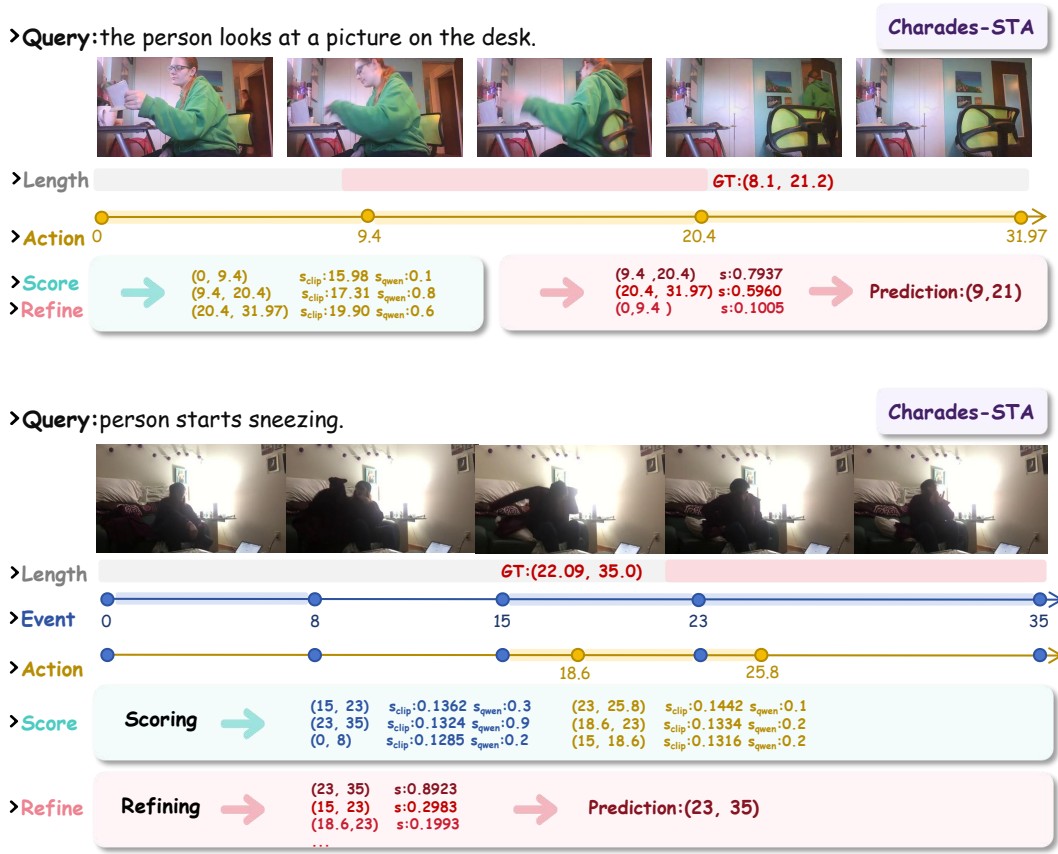

Figure 6: Qualitative examples on short-video datasets. The top case illustrates the result obtained using only the **action-level** decomposition, while the bottom case shows a comparison between **event-level** and **action-level** results.

## A.2 MAIN ALGORITHMS.

### A.2.1 HIERARCHICAL MERGING

**Algorithm.** We enforce hierarchical consistency among event, scene, and action boundaries through a top-down merging strategy. The procedure is summarized in Algorithm 1.

---

**Algorithm 1** Hierarchical merging of temporal boundaries

---

1: **Input:** Feature similarity curves $S^{\text{vit}}, S^{\text{flow}}, S^{\text{dino}}$, video duration $T$, tolerance $\alpha$
2: **Output:** Final boundary set $\mathcal{P}^{\text{final}}$
3: Extract candidate boundaries: $\mathcal{C}^{\text{event}} \leftarrow \texttt{find\_minima}(S^{\text{vit}})$, $\mathcal{C}^{\text{scene}} \leftarrow \texttt{pelt}(S^{\text{flow}})$, $\mathcal{C}^{\text{action}} \leftarrow \texttt{pelt}(S^{\text{dino}})$
4: Merge nearby points: $\mathcal{P}^k \leftarrow \texttt{merge\_numbers}(\mathcal{C}^k, T)$, $k \in \{\text{event, scene, action}\}$
5: Hierarchical merging: $\mathcal{P}^{\text{scene}} \leftarrow \mathcal{M}(\mathcal{P}^{\text{event}}, \mathcal{P}^{\text{scene}}, \alpha)$
    $\mathcal{P}^{\text{action}} \leftarrow \mathcal{M}(\mathcal{P}^{\text{scene}}, \mathcal{P}^{\text{action}}, \alpha)$
6: **Return** $\mathcal{P}^{\text{final}} = \{\mathcal{P}^{\text{event}}, \mathcal{P}^{\text{scene}}, \mathcal{P}^{\text{action}}\}$

---

**Boundary Extraction Functions.** The function $\texttt{find\_minima}$ selects local minima from a similarity curve to generate candidate boundaries, while $\texttt{pelt}$ implements the PELT change point detection algorithm, which identifies temporal transitions based on curve statistics. The function $\texttt{merge\_numbers}$ enforces a minimum distance between boundary points by merging those that are overly close. The hierarchical merging function $\mathcal{M}(\cdot)$ integrates boundaries across levels: for

each higher-level boundary, the closest lower-level point is replaced if within a temporal tolerance $\alpha$, otherwise the higher-level point is inserted.

**Rationale for Multi-Granular Boundary Detection.** Different granularity levels correspond to distinct temporal dynamics. Event-level boundaries reflect long-term semantic shifts and can be reliably captured using local minima in VideoCLIP-based similarity curves. Scene- and action-level boundaries represent shorter and more frequent transitions; for these, PELT efficiently detects multiple change points by modeling curve statistics rather than relying solely on local extrema. This strategy balances computational efficiency with boundary accuracy.

**Hierarchical Merging and Adaptation.** Candidate boundaries are initially extracted independently from each feature space (CLIP-ViT, RAFT flow, DINOv2). The hierarchical merging function $\mathcal{M}(\cdot)$ then integrates higher-level boundaries into lower levels, ensuring fine-grained action segments align with broader temporal structures. In long videos, this merging is essential to maintain global coherence and prevent error accumulation. In short videos, however, hierarchical merging is disabled: the smaller number of segments and sufficient diversity across feature spaces make independent candidates preferable. This design choice, empirically validated in Table 8, allows the framework to adapt to different video lengths while preserving both precision and robustness in temporal localization.

### A.2.2 CANDIDATE REFINEMENT

---
**Algorithm 2** Candidate refinement via Progressive Merging

---
1: **Input:** Initial candidate segments $\{c_i\}$ with scores $\{s_i\}$, score threshold $\theta$
2: **Output:** Refined candidate set $\mathcal{S}_{\text{final}}$ with updated scores
3: **while** merging is possible **do**
4:     **for** each pair $(c_i, c_j)$ not yet merged **do**
5:         **if** $|s_i - s_j| < \theta$ **and** $c_i, c_j$ are temporally proximate **then**
6:             Compute weights $a = s_i/(s_i + s_j)$, $b = s_j/(s_i + s_j)$
7:             Update boundaries of merged segment by score-weighted averaging
8:             Assign merged score $s_{\text{new}} = a \cdot s_i + b \cdot s_j$
9:             Replace $(c_i, c_j)$ with new merged segment
10:         **end if**
11:     **end for**
12: **end while**
13: Rank refined candidates $\mathcal{S}_{\text{final}}$ by updated scores
14: **return** $\mathcal{S}_{\text{final}}$

---

The candidate refinement stage addresses two key challenges arising from hierarchical temporal decomposition (HTD) and the limitations of off-the-shelf VLMs. First, HTD-generated segments may not align perfectly with every video's inherent structure, and query-relevant content often spans multiple hierarchical levels. The initial overlapping candidates from different hierarchical levels may require consolidation to produce temporally coherent predictions. Second, short segments may lack sufficient visual context for reliable VLM assessment, leading to under-scoring, while long segments may dilute salient information with irrelevant content—a known weakness of VLMs in long-video understanding.

To mitigate these issues, we introduce a merging algorithm, merge_segments_all, which consolidates candidates based on score similarity (within a threshold $\theta$) and temporal overlap. The process operates similarly to agglomerative clustering: segments satisfying these criteria are merged, with new boundaries computed as a confidence-weighted average. This design ensures that segments with higher VLM scores exert greater influence on the final localization, thereby correcting for sampling bias and reconciling over-segmentation across hierarchical levels. The final candidate set, $\mathcal{S}_{\text{final}}$, is ranked by a unified score $s_{\text{final}} = \lambda \cdot s_{\text{vlm}} + (1 - \lambda) \cdot s_{\text{clip}}$, balancing cross-modal relevance and temporal coherence in the top-ranked predictions.

### A.2.3 DISTINCTION BETWEEN MERGING STRATEGIES

Our framework incorporates two distinct merging processes: the Hierarchical Merging in the Temporal Decomposition (HTD) stage and the Progressive Merging in the Candidate Refinement (CR) stage. The apparent complexity of having two merging steps is necessary because they serve fundamentally different purposes—one is video-centric and structural, the other is query-centric and semantic.

**HTD Merging (Structural and Video-Centric).** The merging performed in the HTD module (as detailed in Algorithm 1) and Appendix A.2.1) is **query-agnostic** and operates **offline**. Its primary goal is to decompose the video based on its intrinsic visual features (derived from ViT, DINO, RAFT) to mine the inherent temporal hierarchy (actions, scenes, events). The merging process here ensures that all generated candidate segments are **structurally and physically coherent**. These segments act as high-quality, pre-computed **building blocks** for the entire framework. Since HTD is query-independent, it cannot predict the final semantic boundary requested by the user.

**CR Merging (Semantic and Query-Centric).** In contrast, the Progressive Merging in the CR module (in Algorithm 2) and Appendix A.2.2) is **query-centric** and operates **online**. This step relies on the VLM and VideoCLIP similarity scores, representing a deep semantic interaction between the query and the video content. A single natural language query often corresponds to a semantic event that **spans multiple adjacent HTD structural units** (e.g., a "cooking" query may cover HTD's separate "chopping" and "frying" units). The CR merging step dynamically re-aggregates these HTD units that share a high semantic relevance to the specific query, ensuring the output segment captures the complete **semantic scope** of the user's intent.

**Rationale for Two-Stage Merging.** The necessity of the two-stage approach lies in efficiency and accuracy: HTD provides the highly refined, structurally sound components, which reduces the search space significantly. CR then utilizes the powerful VLM to ensure that these structural components are correctly and fully assembled into the **semantically accurate** boundary required by the query. Both stages are indispensable for achieving robust temporal grounding performance.

### A.3 IMPLEMENTATION DETAILS.

This section provides a comprehensive account of the experimental setup, covering feature extraction, model configurations, hyperparameter settings, and dataset-specific strategies. We further include dataset statistics and analysis to contextualize the evaluation and ensure transparency and reproducibility of our results.

### A.3.1 FEATURE EXTRACTION AND SIGNAL CONSTRUCTION

- **Frame Sampling:** Videos are subsampled at a rate of **5 FPS for Charades-STA** and **1 FPS for all other datasets** to balance computational efficiency and temporal resolution.
- **Feature Encoders:** We use frozen, off-the-shelf models:
  - **ViT (CLIP):** ViT-B/32 to extract frame-wise semantic features ($\mathbf{v}_t^{\text{vit}} \in \mathbb{R}^{512}$). Its language-aligned embeddings provide high-level event context when aggregated over several seconds.
  - **DINOv2:** DINOv2-base (ViT-B/14) to extract structural features that are sensitive to scene layout and viewpoint changes, serving as a scene-level signal. Each frame is represented as a 768-dimensional vector ($\mathbf{v}_t^{\text{dino}} \in \mathbb{R}^{768}$).
  - **RAFT**: Optical flow between consecutive frames is extracted using a pre-trained RAFT model (large variant). The magnitude of the flow field is spatially averaged to produce a frame-level motion intensity score ($\mathbf{v}_t^{\text{flow}} \in \mathbb{R}$).
- **Rationale for backbone selection:** We deliberately choose three widely used encoders that emphasize complementary cues at different temporal scales.
  - For *event-level* semantics, we use CLIP ViT-B/32 as it produces image embeddings aligned with natural language and has strong zero-shot transfer, making the aggregated features over several seconds a natural descriptor of high-level activity context.

- For *scene-level* structure, we adopt DINO-v2, whose self-supervised ViT features are sensitive to changes in layout, background, and viewpoint, thus providing a robust signal for shot- and scene-like transitions.

- For *action-level* motion, we employ RAFT-Large, a high-accuracy optical-flow model whose dense flow fields summarize instantaneous motion; the magnitude of the flow directly reflects short, local movements that correspond to action boundaries. Our HTD module only relies on these semantic/structural/motion properties, so other encoders providing similar cues could be substituted without modifying the overall algorithm.

To empirically justify this selection, we conducted an ablation study in Section A.4.3. The results demonstrate that these models are chosen as established representatives with proven generalization capabilities, thereby maximizing the model's performance.

This three-stream design makes HiTeA naturally backbone-agnostic: any encoders that provide comparable event-, scene-, and action-level signals can be plugged into our HTD module without changing the rest of the framework.

### A.3.2 HIERARCHICAL TEMPORAL DECOMPOSITION

- **Boundary Detection:**

  - **Event-level ($\mathcal{C}^{\text{event}}$):** The video is first partitioned into coarse segments of uniform duration. Within each segment, local minima in the $s^{\text{event}}$ curve are identified as potential event boundaries, capturing major semantic shifts.

  - **Scene-level ($\mathcal{C}^{\text{scene}}$) & Action-level ($\mathcal{C}^{\text{action}}$):** The PELT change point detection algorithm is applied to the $s^{\text{scene}}$ and $s^{\text{action}}$ curves, using a radial basis function (RBF) kernel. The penalty term is empirically set to 5 for both feature types to balance sensitivity and robustness in transition detection.

- **Parameters in Hierarchical Merging:**

  - **Merge numbers**: Ensures a minimum distance of 2 seconds between consecutive boundaries within the same level.

  - **Temporal tolerance:** In function $\mathcal{M}$, the temporal tolerance $\alpha$ is set to 5 seconds. A higher-level boundary replaces the nearest lower-level one if their distance is $< \alpha$, otherwise it is inserted.

  - **Activation:** The Hierarchical Merging (HM) process is **enabled for long-video datasets** (Ego4D-NLQ, TACoS) and **disabled for short-video datasets** (Charades-STA, ActivityNet-Captions). This is based on the empirical analysis in Appendix A.4.2.

### A.3.3 CANDIDATE PROPOSAL AND SCORING

- **VideoCLIP-based Pre-filtering:**

  - **Model:** We use a VideoCLIP model (VideoCLIP-XL (Wang et al., 2024a)) to compute the coarse similarity $s_{\text{clip}}$ between a candidate video segment and the text query.

  - **Segment Merging:** Adjacent segments are merged if their $s_{\text{clip}}$ scores differ by less than $\beta = 0.1$. This threshold was empirically set and applied consistently across all datasets. A detailed analysis of the merging behavior and its impact on different datasets is provided in the experimental section 4.4.

  - **Top-K Selection:** The top $k = 3$ segments from each hierarchy level (event, scene, action) are selected for VLM evaluation, resulting in a maximum of 9 segments per video.

- **VLM-based Scoring:**

  - **Model:** We utilize Qwen2.5-VL (7B) as our primary frozen VLM for its strong video-language alignment capability.

  - **Scoring:** The model computes a semantic similarity score $s_{\text{vlm}} \in [0, 1]$ for each candidate segment. The final score for a candidate is computed as $s_{\text{final}} = \lambda \cdot s_{\text{vlm}} + (1 - \lambda) \cdot s_{\text{clip}}$, with $\lambda = 0.99$. We set $\lambda = 0.99$ to ensure that the final ranking is dominated by the VLM's semantic score, while still leveraging CLIP as a lightweight tie-breaker.

Table 6: Dataset statistics and characteristics. Key attributes include the number of videos, number of language queries, average video length, average moment length, annotation source, and domain.

| Benchmark | #Videos | #Queries | Video Len. (s) | Moment Len. (s) | Domain |
|---|---|---|---|---|---|
| Charades-STA | 1.3k | 3.7k | 29 | 7.8 | Activity |
| ANet-Captions | 4.9k | 17.1k | 118 | 40.2 | Activity |
| TACoS | 25 | 4.0k | 368 | 31.9 | Cooking |
| Ego4D-NLQ | 415 | 4.6k | 472 | 10.7 | Ego-centric |

In practice, the VLM often assigns identical or nearly identical scores to multiple candidate segments, making it difficult to establish a clear ranking. Incorporating a small weight from CLIP provides additional discriminative power in such cases, while the near-unit weight on the VLM score prevents CLIP from interfering with the primary semantic alignment.

### A.3.4 CANDIDATE REFINEMENT

- **Merging Criteria:** Temporally proximate or overlapping segments are iteratively merged if they satisfy both conditions: (1) their boundaries are within **1 second** of each other, and (2) their unified confidence scores $s_{\text{final}}$ differ by less than **0.14**.
- **Boundary Update:** New boundaries for merged segments are computed as the score-weighted average of the original boundaries.

### A.3.5 DATASET ANALYSIS

We conduct experiments on four widely used benchmarks: Charades-STA, ActivityNet-Captions, TACoS, and Ego4D-NLQ. As summarized in Table 6, these datasets vary significantly in scale, video duration, and domain coverage, thereby providing a comprehensive evaluation setting. Charades-STA and ActivityNet-Captions consist of short-to-medium length activity videos paired with natural language queries, while TACoS focuses on long cooking videos with fine-grained temporal annotations. Ego4D-NLQ, in contrast, represents an egocentric perspective with long untrimmed videos and query annotations grounded in first-person interactions. Importantly, all four datasets are annotated by human annotators, ensuring high-quality language descriptions and reliable temporal grounding labels. This diversity in domain, video duration, and query density makes the set of benchmarks complementary and well-suited for evaluating the scalability and generalization ability of temporal grounding methods.

Figure 7 further illustrates the distribution of normalized moment lengths across the four datasets. Charades-STA exhibits relatively short moments, mostly concentrated within 0.1–0.3 of the video duration, reflecting its focus on fine-grained activity descriptions. ActivityNet-Captions displays a broader spread, but still biases toward shorter segments, which is consistent with the open-domain nature of its videos and captions. TACoS, in contrast, shows a significant proportion of long moments, often covering more than half of the video duration, highlighting the dense and procedural structure of cooking videos. Ego4D-NLQ presents a more balanced distribution, with moments spanning a wide range of relative lengths, reflecting the egocentric setting where queries may correspond to both brief interactions and extended activities.

These distinct distributions underscore the heterogeneity of temporal grounding benchmarks. Datasets with predominantly short moments (e.g., Charades-STA) emphasize precise boundary detection, while those with longer or more varied segments (e.g., TACoS, Ego4D-NLQ) challenge methods to handle diverse temporal scales. This diversity motivates the need for our hierarchical decomposition and adaptive scoring framework, which is explicitly designed to cope with the variability in moment granularity across benchmarks.

**Dataset-Specific Configurations.** Most hyperparameters remain fixed across datasets. The dataset-specific adjustments are the frame sampling rate and the activation of the Hierarchical Merging module, as detailed above. This demonstrates the robustness and generalizability of our framework.

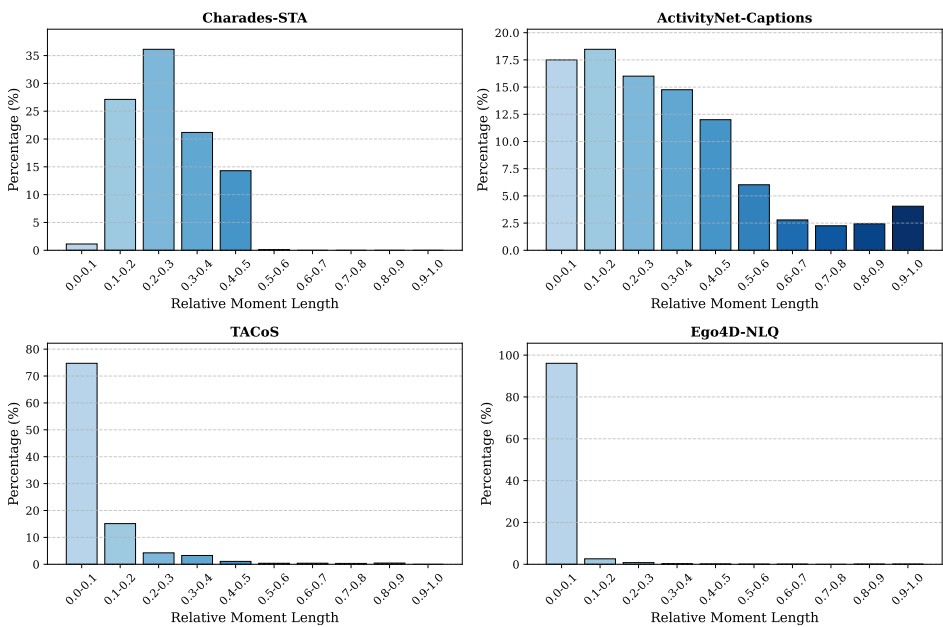

Figure 7: Distribution of normalized moment lengths across four benchmarks. The x-axis denotes the ratio of the annotated moment length to the corresponding video duration, while the y-axis shows the percentage of queries in each bin.

Table 7: Ablation Study on Visual-Language Backbones.

| Model | Size | TACoS | | | | Charades-STA | | | |
|---|---|---|---|---|---|---|---|---|---|
| | | 0.1 | 0.3 | 0.5 | mIoU | 0.3 | 0.5 | 0.7 | mIoU |
| Videoclip-XL | 0.6B | 22.19 | 12.95 | 7.38 | 9.50 | 55.03 | 38.23 | 20.27 | 37.17 |
| +InternVideo2.5 | 7B | 40.94 | 24.50 | 13.24 | 17.03 | 69.11 | 48.39 | 25.24 | 45.94 |
| +Qwen2.5-VL | 3B | 41.85 | 22.66 | 11.61 | 16.36 | 62.15 | 42.39 | 22.20 | 41.50 |
| **+Qwen2.5-VL** | **7B** | **44.94** | **29.08** | **16.10** | **19.79** | **69.62** | **48.52** | **25.59** | **46.29** |

## A.4 SUPPLEMENTARY EXPERIMENTS.

### A.4.1 IMPACT OF VISUAL-LANGUAGE MODEL SELECTION

The results in Table 7 reveal several important insights regarding the role of different VLMs within the HiTeA framework. When using only Videoclip-XL without a large-scale VLM, performance on short-video datasets remains reasonable, but results on long videos are significantly lower. This suggests that effective long-video grounding requires the strong generalization capabilities provided by large VLMs. Across models, performance generally scales with model capacity: as the number of parameters increases from 3B to 7B in Qwen2.5-VL, metrics on both TACoS and Charades-STA improve consistently. This trend indicates that larger VLMs offer stronger visual-language alignment, enabling more accurate evaluation of the semantic relevance between video segments and text queries.

Equally notable is that VLMs of similar scale demonstrate comparable performance. For example, Qwen2.5-VL and InternVideo2.5, both with 7B parameters, achieve nearly identical results on Charades-STA. This suggests that, provided the model has sufficiently rich general representations, HiTeA can leverage different backbone VLMs without any task-specific adjustment or fine-tuning. Overall, these observations highlight the framework's robustness and generality, confirming that the combination of hierarchical temporal decomposition and large pre-trained VLMs can consistently support effective long-video temporal grounding in a fully training-free setting.

Table 8: Impact of Hierarchical Merging.

| Method | Charades-STA | | | | ANet-Captions | | | |
|---|---|---|---|---|---|---|---|---|
| | 0.3 | 0.5 | 0.7 | mIoU | 0.3 | 0.5 | 0.7 | mIoU |
| **w HM** | 69.65 | 44.30 | 22.66 | 45.09 | 53.65 | 30.55 | 16.24 | 37.30 |
| **w/o HM** | 69.62 | 48.52 | 25.59 | 46.29 | 54.46 | 31.1 | 16.57 | 37.93 |

### A.4.2 IMPACT OF HIERARCHICAL MERGING.

The results in Table 8 reveal a nuanced but consistent pattern: the application of our Hierarchical Merging (HM) module leads to a slight performance degradation on short-video datasets (Charades-STA and ANet-Captions), while its benefits are pronounced and essential for long videos (as established in our main results). We argue this is not a shortcoming but a critical, intentional design feature that demonstrates our framework's adaptive capability.

The core rationale lies in the fundamental structural difference between short and long videos. Short videos (e.g., Charades-STA) typically contain fewer semantic segments and a less complex hierarchical temporal structure. In this setting, the candidate boundaries extracted from the three complementary feature spaces (event, scene, action) are already sparse and highly informative. Enforcing hierarchical merging here can be overly restrictive; it risks diluting the valuable diversity of these independent signals by forcing them into a single hierarchy. The marginally superior performance **without HM** (w/o HM) suggests that for short videos, preserving this diversity of boundary hypotheses is more beneficial than imposing a rigid top-down structure. The different feature spaces effectively provide an ensemble of perspectives, leading to richer and more accurate localization.

Conversely, long videos exhibit a deeply nested temporal hierarchy with abundant redundant content. Without a mechanism to enforce consistency across different levels of granularity, the sheer number of potential boundaries from various features would lead to temporal clutter, error accumulation, and a loss of global coherence. The HM module is indispensable here. It acts as a regularizer, pruning spurious detections and aligning fine-grained action transitions with broader scene and event contexts. This creates a clean, coherent, and multi-scale temporal scaffold that is crucial for efficiently and accurately navigating long durations.

Therefore, our decision to disable HM for short videos and enable it for long videos is a principled one, grounded in the data's inherent characteristics. It demonstrates that our framework is not a one-size-fits-all solution but intelligently adapts its strategy to the input. This adaptive design is a strength, contributing to robust and state-of-the-art performance across both short and long-form video benchmarks.

### A.4.3 IMPACT OF FEATURE EXTRACTION MODEL SELECTION.

To investigate the impact of specific feature extractors and verify the generalizability of our framework, we conducted ablation studies by replacing the backbone models at the Action and Event levels.

Specifically, we replaced the RAFT optical flow model with FlowNet (Dosovitskiy et al., 2015) for the Action level, replaced DINOv2 with iBOT (Zhou et al., 2021) for the Scene level, and replaced the ViT-CLIP model with a standard Swin-Transformer (Liu et al., 2021) for the Event level. The results on Charades-STA are reported in Table 9.

As observed in Table 9, while the framework maintains competitive performance using alternative backbones, our default configuration yields the highest accuracy.

**Action Level**: Using FlowNet results in a performance drop of 1.27% mIoU. This validates our choice of RAFT, as its dense, pixel-level flow estimation captures fine-grained motion boundaries more effectively than the coarser features from FlowNet.

**Scene Level**: Replacing DINOv2 with iBOT leads to a 1.33% decrease in mIoU. While iBOT utilizes Masked Image Modeling (MIM) to effectively capture local structural details—crucial for detecting scene transitions—DINOv2 achieves superior results. This justifies our selection of DI-

Table 9: Impact of Feature Extraction Model Selection.

| Model | Level | Charades-STA | | | |
|---|---|---|---|---|---|
| | | 0.3 | 0.5 | 0.7 | mIoU |
| Flownet(Dosovitskiy et al., 2015) | Action | 68.92 | 46.94 | 23.41 | 45.02 |
| iBOT(Zhou et al., 2021) | Scene | 69.14 | 46.37 | 23.47 | 44.96 |
| Swin-Transformer(Liu et al., 2021) | Event | 67.45 | 45.46 | 23.76 | 44.47 |
| **Ours** | | **69.62** | **48.52** | **25.59** | **46.29** |

NOv2, as it inherits the structural awareness of iBOT while benefiting from larger-scale curated data and optimized training recipes, resulting in more robust generalized features.

**Event Level**: Replacing ViT with Swin-Transformer leads to a 1.82% drop in mIoU. This confirms the necessity of Vision-Language alignment at the top level. ViT explicitly aligns visual concepts with language queries, allowing for more accurate detection of high-level semantic narrative shifts compared to purely visual features.

These empirical results substantiate that our core contribution lies in the hierarchical processing strategy itself (Event-Scene-Action), rather than being tied to specific architectures. However, our default configuration yields the highest accuracy. This demonstrates that while the hierarchical structure is the primary driver of success, our selection of ViT, DINO, and RAFT—chosen as established representatives with proven generalization capabilities—is essential for maximizing the model's overall potential, thereby empirically justifying our architectural decisions.

### A.5 COMPUTATIONAL COMPLEXITY

HiTeA consists of two major stages: hierarchical candidate generation and VLM-based scoring.

**1. Hierarchical Candidate Generation.** Given a video with $N$ frames, we first extract multi-scale VideoClip features and apply the Hierarchical Temporal Decomposition (HTD) module to generate candidate segments at multiple temporal granularities. The number of frames $N$ is typically very large for long videos. The computation involved in this stage primarily includes feature extraction, lightweight aggregation, and candidate filtering, which scale approximately linearly with the video length:

$$\mathcal{O}_{\text{HTD}} = \mathcal{O}(H \cdot N) \, , \tag{5}$$

where $H = 3$ is the number of hierarchical levels. Since these operations do not involve heavy neural network inference, this stage is computationally efficient even for long videos.

**2. VLM-Based Scoring.** The computationally expensive component is the scoring of candidate segments using a frozen vision–language model (VLM). HiTeA applies a top-$K$ filtering strategy at each hierarchical level, such that only the most promising candidates are evaluated by the VLM. Let $K_i$ denote the number of candidates retained at level $i$, then the total number of VLM evaluations is:

$$N_{\text{VLM}} = \sum_{i=1}^{H} K_i \, . \tag{6}$$

The corresponding complexity is:

$$\mathcal{O}_{\text{VLM}} = \mathcal{O}\left(\sum_{i=1}^{H} K_i\right) \approx \mathcal{O}(H \cdot K) \, , \tag{7}$$

where $K \ll N$ ensures that VLM inference is tractable, even for very long videos. This represents a substantial reduction compared to a brute-force search over all possible start-end pairs, which would require $O(N^2)$ VLM evaluations.

## A.6 SCALABILITY AND RUNTIME ANALYSIS

**Clarification on Multi-Model Usage.** Although HiTeA leverages five pre-trained components to ensure robust performance, most of them are used strictly in an **offline preprocessing stage**. The pipeline is structurally decoupled as follows:

- **Offline preprocessing (executed once per video):** (1) a video encoder (e.g., ViT) for event-level features; (2) a self-supervised backbone (e.g., DINO) for complementary scene-level cues; (3) a motion model (e.g., RAFT) for optical-flow features. These backbones are only used for feature extraction and do not participate in online reasoning.

- **Online interaction (executed per query):** (4) a lightweight retriever (e.g., VideoCLIP) for candidate selection; (5) a VLM (e.g., Qwen2.5-VL) for high-level reasoning over a small set of candidates.

During user interaction, only components (4) and (5) are invoked. In typical deployments, video-level features are pre-computed and cached, so the heavy perceptual cost is amortized across queries and does not affect per-query latency. Moreover, the three offline backbones (ViT, DINO, RAFT) are fully parallelizable across GPUs.

**Computational Complexity.** We compare HiTeA against three representative training-free baselines: the dense-scanning TFVTG (Zheng et al., 2024), the compositional proposal method of Luo et al. (2024), the BLIP-based zero-shot VMR (Wattasseril et al., 2023), and the dense-captioning approach Moment-GPT (Xu et al., 2025). To rigorously evaluate scalability, we conduct experiments on QVHighlights and TACoS, which are characterized by distinct temporal distributions with average video lengths of $\sim$150s and $\sim$368s, respectively.

- **Baselines ($O(L^2)$ or $O(L)$).** TFVTG requires dense proposal enumeration over all temporal pairs, leading to $O(L^2)$ complexity in the video length $L$. Luo et al. (2024) operate on a fixed number of video snippets $L_v$ (32 in their implementation) and decompose each query into $L_Q$ simple queries; their online cost therefore scales as $O(L_v \cdot L_Q)$ and is effectively insensitive to the raw video length, but the original paper does not report wall-clock latency. The BLIP-based zero-shot VMR approach (Wattasseril et al., 2023) performs sparse per-second BLIP/BLIP-2 scoring followed by watershed-style merging, resulting in an $O(L)$ online cost in $L$ with a lighter per-frame model than full MLLMs, but again without reported runtime on long-video benchmarks. Moment-GPT performs frame-by-frame MLLM captioning, which scales linearly with $L$ ($O(L)$) but with a substantial constant factor due to the heavy MLLM.

- **HiTeA ($O(K)$ online).** HiTeA first abstracts each video into a small set of structural events via Hierarchical Temporal Decomposition (HTD), and then selects a fixed number of candidates (Top-$K$) for VLM reasoning. The online reasoning complexity is thus reduced to $O(K)$, where $K$ is the number of selected candidates. We set $K{=}9$ by default, making the online cost effectively insensitive to the raw video length $L$ in our regime of interest.

**Practical Runtime Decomposition.** To make the computational cost more transparent, we further decompose HiTeA's runtime into two stages.

**Stage 1: Offline feature extraction (once per video).** The first three backbones in HiTeA—ViT, DINO, and RAFT—are used solely for feature extraction. In practical deployments, video features (e.g., RGB embeddings, optical flow, CLIP-style embeddings) are typically pre-computed and stored. These three models can be executed fully in parallel. On TACOS videos, extracting all features for a single video takes about **6–7s**, dominated by RAFT optical-flow computation. If we disable optical flow and only keep event/scene features, this offline stage reduces to **1–2s**, confirming that RAFT is the main contributor to offline cost.

**Stage 2: Online per-query reasoning (user-perceived latency).** This stage determines the latency observed by the end user. It involves two models (VideoCLIP and the VLM) and two lightweight algorithms (HTD and CR):

- *Algorithmic overhead is negligible.* HTD operates on cached features and runs in approximately **0.0012s** per video on average; CR requires only **0.0001s** per query in our measurements. Their contribution to total latency is negligible compared to VLM inference.

- *Cascade-filter design reduces VLM calls.* A straightforward zero-shot design would apply a VLM to all (or densely sampled) segments, leading to hundreds of VLM inferences on long videos. HiTeA instead adopts a two-stage cascade: (i) a very fast VideoCLIP module scores all HTD candidates (e.g., $\sim$38 on TACOS and $\sim$147 on Ego4D), and (ii) only the Top-$K$ candidates ($K{=}9$) are passed to the heavy VLM. In our implementation, a single Qwen2.5-VL forward pass takes about 1s with an optimized runtime, so the overall VLM cost is tightly controlled.

Table 10: Runtime efficiency breakdown. **Offline cost** is dominated by RAFT (optical flow) but is amortized over queries. **Online latency** remains approximately constant because the heavy VLM is restricted to Top-$K$ candidates, avoiding the length-dependent scaling observed in dense captioning approaches. Most training-free works do not report time anaylsis, so we primarily list their online complexity. For Moment-GPT, we report the QVHighlights latency from (Xu et al., 2025) and provide a coarse extrapolation on TACOS based on average video duration.

| Method | Mechanism | Online Complexity | Inference Latency (s / query) | |
|---|---|---|---|---|
| | | | **Short (QVH)** | **Long (TACOS)** |
| **BLIP-VMR** (Wattasseril et al., 2023) | Sparse 1s scoring + merging | $O(L) \times$ BLIP | 12.7 | 25 (est.) |
| **TFVTG** (Zheng et al., 2024) | Dense scanning | $O(L^2)$ | - | - |
| Luo et al. (2024) | Cartesian product | $O(L_v \cdot L_Q)$ | - | - |
| **Moment-GPT** (Xu et al., 2025) | Dense frame captioning | $O(L) \times$ MLLM | 16.1 | 32 (est.) |
| **HiTeA (Ours)** | **Cascade reasoning** | $\mathbf{O(K)}, K{=}9$ | **See below** | **See below** |
| *— HiTeA breakdown: offline vs. online —* | | | | |
| **1. Offline cost** | ViT + DINO + RAFT | $O(L)$ | 3.0–4.0 | 6.0–7.0$^\dagger$ |
| **2. Online latency** | **VideoCLIP filter $\to$ VLM** | $\mathbf{O(K)}$ | 7.0–8.0 | 7.0–8.0 |

$^\dagger$ Dominant cost is RAFT ($\sim$5–6s). Without flow, offline cost drops to $\sim$1–2s. HTD overhead is negligible ($\sim$0.0012s).

For Luo et al. (2024), $L_v$ denotes the fixed number of video snippets (32 in their implementation) and $L_Q$ is the number of simple queries.

The inference latencies on TACOS for BLIP-VMR and Moment-GPT are estimated based on average video length.

On TACOS, the average per-query scoring time of HiTeA is **7–8s**. Empirically, most of the time is spent on the Qwen2.5-VL inferences, while VideoCLIP and the HTD/CR computations contribute only a tiny fraction. In other words, HiTeA introduces just enough structure (HTD + cascade filtering) to eliminate the hundreds of redundant VLM calls that a naive "one-VLM-over-all-segments" design would incur on long videos.

**Summary.** Table 10 and the above breakdown show that while offline preprocessing naturally scales with video length (e.g., 3–4s $\to$ 6–7s from QVHighlights to TACOS), HiTeA's **online latency** remains stable at approximately 7.5s per query. This confirms that our sparse structural reasoning design effectively shields the user experience from the computational heaviness of the five underlying models, and maintains a practical efficiency–accuracy trade-off even on long videos.

### A.7  LIMITATIONS AND FUTURE DIRECTIONS.

The primary limitation of HiTeA stems from its core design: the hierarchical decomposition delivers maximal benefit for long, untrimmed videos with complex temporal structures. On very short clips, the advantages of a strict multi-level hierarchy are naturally diminished. Our ablation studies, however, reveal that comparable performance can be achieved with a simplified one- or two-level partitioning, indicating that a fixed three-layer hierarchy is not always necessary.

Furthermore, the decomposition need not be strict in practice. The optimal granularity may vary depending on factors such as video domain, length, and even the specific query, leading to organic cross-level overlaps and merges. This inherent complexity is precisely what our Candidate Refinement module is designed to handle. The framework's adaptability points toward promising future directions. For instance, rather than relying on a fixed hierarchy, a dynamic strategy could be developed to adaptively determine the optimal decomposition depth based on video properties. More fun-

damentally, alternative strategies for temporal segmentation itself—beyond the action-scene-event paradigm we adopted—are worth exploring. The principle of multi-scale temporal analysis could also be generalized to other video understanding tasks, such as dense video captioning or summarization. This work serves as an initial step in this direction, establishing a foundation for future research into more flexible and general temporal reasoning models.

### A.8 THE USE OF LLM.

We clarify that a large language model (LLM) was used solely as a writing assistant in the preparation of this paper. Specifically, the LLM was employed to polish the text, improve grammar, and refine the clarity and flow of exposition. It was not involved in the design of methods, implementation of algorithms, execution of experiments, or analysis of results. All technical contributions, experiments, and conclusions presented in this work are entirely the authors' own.

