# OpenReview forum: "HiTeA: Hierarchical Temporal Alignment for Training-Free Long-Video Temporal Grounding"
_ICLR.cc/2026/Conference — ICLR 2026 Poster_

### Official Review · Reviewer_RwT5 · 2025-10-26

**Soundness:** 2
**Presentation:** 3
**Contribution:** 2
**Rating:** 4
**Confidence:** 4

**Summary:**

This paper introduces HiTeA, a training-free framework for temporal grounding in untrimmed videos. The core of HiTeA is its Hierarchical Temporal Decomposition (HTD) mechanism, which structures a video into multi-scale temporal units (events, scenes, actions). It leverages pre-trained vision-language models to match text queries with candidate segments without any task-specific training and employs a refinement module to generate final predictions. The authors report extensive experiments on Charades-STA, ActivityNet-Captions, Ego4D-NLQ, and TACoS datasets.

**Strengths:**

1. The manuscript is well-written and clearly structured.

2. The supporting figures, particularly the conceptual diagrams, are well-designed and effectively aid in understanding the proposed framework.

**Weaknesses:**

1. The proposed hierarchical relationship among events, scenes, and actions appears to be somewhat problematical and lacks a strong empirical or theoretical justification. For instance, in real-world videos, the relationship can be more complex and non-hierarchical; a single scene might encompass multiple distinct events. A more detailed rationale or citation for this specific structural choice would strengthen the methodological foundation.

2. The experimental evaluation would benefit from a direct comparison with other recent, state-of-the-art training-free methods for temporal grounding or related video-language tasks. [a] Universal Video Temporal Grounding with Generative Multi-modal Large Language Models, arXiv 2025.  [b] Zero-Shot Video Grounding With Pseudo Query Lookup and Verification, TIP 2021.

3. In the ablation study of Table 3, the performance improvements from the individual components of the Hierarchical Temporal Decomposition (HTD) on the TACoS dataset seem relatively marginal, and in some configurations, a component even appears to degrade performance. The manuscript should provide an in-depth analysis discussing the potential reasons for this observed instability or limited impact. Furthermore, the ablation study appears incomplete as it does not present the results of using the Candidate Refinement module in isolation (e.g., on uniformly sampled segments). Showing this baseline is critical to disentangle and quantify its specific contribution to the overall performance.

4. The generalization and robustness of the method could be more convincingly demonstrated by including evaluations on the QVHighlights dataset. A key challenge of this dataset is that a single query can correspond to multiple moments. Testing on QVHighlights would provide valuable insights into the model's capability to handle such complex, multi-instance grounding scenarios.

5. The justification for the specific choice of pre-trained models for feature extraction (ViT-B/32 for events, DINO-v2 for scenes, RAFT-Large for actions) is insufficient. The authors should clarify the specific properties or prior evidence that make these particular models representative of their respective hierarchical levels. A discussion on whether alternative models were considered and why these were chosen would strengthen the methodological design.

6. The inference cost of utilizing five different pre-trained models is a significant practical concern, especially for processing long videos. The manuscript should include a discussion, and ideally a quantitative analysis, of the computational efficiency and inference speed. This is a critical factor for real-world applicability and should be transparently reported.

**Questions:**

Please refer to the weaknesses.

---

> ### Author Response · Authors · 2025-11-21
> **Summarize**
>
> We sincerely thank the reviewer for the comprehensive evaluation and the encouraging feedback on our manuscript’s **clarity and effective visualization**. We have carefully addressed the concerns regarding the hierarchical rationale, baselines, and efficiency as follows:
>
> - **Response to W1 (Hierarchy Rationale)**: We clarify that the "Event-Scene-Action" structure serves as an **initial scaffold** rather than a rigid rule. We emphasize that the **Candidate Refinement (CR)** module explicitly handles non-hierarchical complexities.
>
> - **Response to W2 (Baselines)**: We explicitly distinguish HiTeA’s **strictly parameter-frozen** paradigm from the optimization-based "zero-shot transfer" methods (UniTime, Lu et al.), demonstrating that HiTeA achieves superior or competitive performance without any training cost.
>
> - **Response to W3 (Ablation Analysis)**: We provide the requested **"Uniform+CR" baseline** and clarify that the perceived "instability" is an artifact of the loose R1@0.1 metric on long videos. Stricter metrics confirm the consistent contribution of HTD.
>
> - **Response to W4 (QVHighlights)**: We added experiments on **QVHighlights**, achieving **SOTA zero-shot performance (60.67% mAP)**, demonstrating robust multi-instance grounding.
>
> - **Response to W5 & W6 (Models & Efficiency)**: We justified the model choices (mapping explicitly to Motion/Structure/Semantics) and provided a **runtime analysis** (App. A.6), showing that HiTeA maintains a **highly efficient, length-invariant online** latency (~7-8s).

---

> ### Author Response · Authors · 2025-11-21
> **1. For Weakness 1**
>
> We sincerely thank the reviewer for highlighting this important point. We fully acknowledge that real-world video relationships can indeed be more complex and non-hierarchical, where a single scene might encompass multiple distinct events. We would like to clarify that the Event–Scene–Action paradigm in our work is not intended to be a rigid, universally applicable hierarchy, but rather a structured, multi-scale abstraction grounded in established cognitive and video analysis literature.
>
> **1.1 Theoretical and Empirical Rationale for the Multi-Scale Structure**
>
> Our Event–Scene–Action framework is a multi-scale abstraction designed to integrate three widely recognized foundational units of video comprehension, which collectively provide a robust framework for temporal grounding.
>
> - **Action Segmentation**: Focuses on the lowest temporal granularity, capturing fine-grained physical movements (e.g., chopping, stirring). This is a well-established unit of activity recognition ([a] Combining embedded accelerometers with computer vision for recognizing food preparation activities. In Proceedings of the 2013 ACM International Joint Conference on Pervasive and Ubiquitous Computing, pages 729–738, 2013.).
>
> - **Scene Detection**: Represents the mid-level granularity, typically marking structural or contextual shifts (e.g., changes in camera angles or scene locations). This is a core concept in video structuring ([b]Shot contrastive self supervised learning for scene boundary detection. In Proceedings of the IEEE/CVF Conference on Computer Vision and Pattern Recognition, pages 9796–9805, 2021.).
>
> - **Event Segmentation**: Captures the coarse-grained level, detecting high-level semantic themes (e.g., "cooking," "talking") that encapsulate coherent activities ([c] Generic event boundary detection: A benchmark for event segmentation. In Proceedings of the IEEE/CVF International Conference on Computer Vision, 2021.).
>
> The Hierarchical Temporal Decomposition (HTD) module is designed to efficiently incorporate these three foundational granularities into an initial, structured segmentation, which serves as a pragmatic starting point for semantic retrieval.
>
> **1.2 Explicit Non-Hierarchy and Flexibility in the Framework**
>
> We agree that real-world data is often non-hierarchical. Our complete system, HiTeA, is explicitly designed to overcome the limitations of a strict hierarchy through the interaction of its two main modules:
>
> - **HTD's Role (Structure)**: Provides the initial segmentation based on cognitive cues (Action $\to$ Scene $\to$ Event) for computational efficiency.
>
> - **Candidate Refinement (CR) Module's Role (Flexibility)**: This module is crucial for handling non-hierarchical complexity. The CR module dynamically refines the segmented candidates based on the query's semantics, rather than strictly enforcing the fixed boundaries set by the HTD.
>
>   - **Example**: If a query (e.g., "the entire cooking process") spans multiple scene transitions or event boundaries, the CR module will merge the initial segments into a single, semantically coherent unit, effectively bypassing the rigid hierarchy established by the HTD.
>
> This combination ensures that while the HTD module offers an efficient, structured representation, the CR module introduces the necessary flexibility and adaptability to align with the non-hierarchical, complex nature of real-world video data, as demonstrated by our results on TACoS and Charades-STA.

---

> ### Author Response · Authors · 2025-11-21
> **2. For Weakness 2**
>
> We thank the reviewer for pointing out these relevant state-of-the-art works. We have carefully studied both [a] UniTime-Zero and [b] Lu et al. and have included them in our revised Tab. 1 and Tab. 2. Below, we discuss methodological distinctions and performance gaps.
>
> **2.1 Methodological Distinction:  Fully Training-Free vs. Zero-Shot Transfer**
>
> It is crucial to distinguish the strict "Fully Training-Free" setting of HiTeA from the "Zero-Shot Transfer" or "Pseudo-Supervision" settings used in the cited works.
>
> - **HiTeA (Ours)**: Our framework operates under a strictly **parameter-frozen** paradigm. It utilizes off-the-shelf pre-trained components (e.g., Qwen2.5-VL, VideoCLIP) and performs inference directly **without any gradient updates, fine-tuning, or optimization** on either source or target datasets.
>
> - **Cited Methods (Optimization-based)**: In contrast, the cited methods involve a significant training phase involving parameter optimization:
>
>   - **[a] UniTime-Zero (arXiv 2025)**: As detailed in their Sec. 2.3, this method adopts a Zero-Shot **Transfer** setting. It requires an extra training phase on large-scale external video datasets before being applied zero-shot to the target benchmark.
>
>   - **[b] Lu et al. (TIP 2021)**: This method relies on a **Pseudo-Supervision** paradigm (Sec. III.E), where a dedicated Video Grounding model is trained and optimized using generated pseudo-labels.
>
> **2.2 Empirical Comparison and Superiority**
>
> Despite HiTeA operating under a significantly more restrictive constraint (i.e., no training cost), we compare it against these optimization-based methods in the revised paper. As shown in the updated tables, HiTeA exhibits superior or highly competitive performance:
>
> - **Short-Video Benchmarks (SOTA Performance)**: HiTeA significantly outperforms the cited methods. On Charades-STA, we achieve a new state-of-the-art for the zero-shot paradigm (e.g., 69.62% R1\@0.3 and 46.29% mIoU). Similarly, on ANet-Captions, HiTeA surpasses competitors across **all key metrics**, improving R1\@0.3 (+5.2%), R1\@0.7 (+1.3%), and mIoU (+4.7%).
>
> - **Long-Video Benchmarks (Competitive Robustness)**: On challenging long-form datasets like TACoS, HiTeA achieves an mIoU of 19.79%. This performance is highly competitive, outperforming or closely trailing methods that benefit from explicit training phases.
>
> **2.3 Final Clarification**:
>
> While [a] and [b] rely on training phases (via external data or pseudo-labels), HiTeA demonstrates that high-performance video grounding can be achieved through a purely inference-based mechanism (HTD + CR). This highlights the efficiency and generalization capability of our approach without the need for task-specific optimization.

---

> ### Author Response · Authors · 2025-11-21
> **3. For Weakness 3**
>
> **3.1 Clarification on "Instability": The Artifact of Loose Metrics**
>
> We have updated Sec. 4 and Tab. 4(revised paper) to include the requested baselines. Regarding the perceived instability on TACoS, we demonstrate that this is an **artifact of the R1\@0.1 metric** rather than model inconsistency.
>
> - **The "Loose Metric" Bias**: On long videos (TACoS), the R1\@0.1 threshold is extremely tolerant. Even coarse, uniform segments often accidentally overlap with the ground truth enough to register a "hit." This creates a ceiling effect where simple baselines appear artificially competitive.
>
> - **True Precision via Stricter Metrics**: When evaluated on boundary-sensitive metrics (R1\@0.5, mIoU), the "instability" vanishes. Our HTD-based variants consistently outperform uniform and single-cue baselines by a significant margin (e.g., mIoU), confirming that HTD produces semantically aligned boundaries rather than accidental overlaps.
>
> **3.2 Isolating the contribution of Candidate Refinement (CR)**
>
> To disentangle CR from HTD, we compared applying CR to uniformly sampled segments ("Uniform + CR") versus our structured segments ("HTD + CR").
>
> - **Uniform (14.87 mIoU) $\rightarrow$ Uniform + CR (16.67 mIoU)**: CR alone yields gains by aggregating semantic information, even on coarse segments.
>
> - **Uniform + CR (16.67 mIoU) $\rightarrow$ HTD + CR (19.79 mIoU)**: The full method achieves the best performance (+3.12 mIoU).
>
> This confirms that structural decomposition (HTD) and semantic refinement (CR) are **complementary**. HTD provides physically meaningful "structural atoms" (events/scenes), while CR performs fine-grained semantic matching. Uniform sampling cannot provide the structural precision required for the highest performance, regardless of the CR module.

---

> ### Author Response · Authors · 2025-11-21
> **4. For Weakness 4**
>
> We appreciate the suggestion to evaluate QVHighlights. This benchmark is indeed critical for assessing robustness due to its "one-query, multiple-moments" nature, which requires models to retrieve **all relevant instances** instead of only the single best one.
> | Method | Sup | R1\@0.5 (Test) | R1\@0.7 (Test) | mAP\@0.5 (Test) | mAP\@avg (Test) | R1\@0.5 (Val) | R1\@0.7 (Val) | mAP\@0.5 (Val) | mAP@avg (Val) |
> |--------|-----|---------------|---------------|----------------|----------------|--------------|--------------|---------------|---------------|
> | **Supervised (FS)** | | | | | | | | | |
> | Moment-DETR | FS | 52.9 | 33.0 | 54.8 | 30.7 | 54.2 | 33.4 | 55.4 | 31.1 |
> | UniVTG | FS | 58.9 | 40.9 | - | - | - | - | - | - |
> | VTimeLLM | FS | 47.2 | 29.3 | 47.3 | 27.4 | 48.8 | 29.5 | 49.3 | 26.8 |
> | Mr.BLIP | FS | 74.8 | 60.5 | 76.1 | 63.4 | - | - | - | - |
> | **Weakly-Supervised (WS)** | | | | | | | | | |
> | CNM | WS | 14.1 | 4.0 | 11.8 | - | - | - | - | - |
> | CPL | WS | 30.8 | 10.8 | 22.8 | - | - | - | - | - |
> | CPI | WS | 32.3 | 11.8 | 23.7 | - | - | - | - | - |
> | **Unsupervised (US)** | | | | | | | | | |
> | PZVMR | US | 14.2 | 4.9 | 15.7 | 4.6 | 12.6 | 5.1 | 16.2 | 5.3 |
> | DSCNet | US | - | - | - | - | 12.3 | 3.5 | 10.4 | 2.7 |
> | **Zero-Shot (ZS)** | | | | | | | | | |
> | UniVTG† | ZS | 25.2 | 9.0 | 27.4 | 10.9 | - | - | - | - |
> | TimeSuite† | ZS | 12.3 | 9.2 | - | - | - | - | - | - |
> | VTimeLLM† | ZS | 26.1 | 11.2 | - | - | - | - | - | - |
> | UniTime-Zero† | ZS | 41.0 | 31.5 | - | - | - | - | - | - |
> | Diwan et al. | ZS | - | - | - | - | 48.3 | 31.0 | 47.3 | 28.0 |
> | Moment-GPT | ZS | 58.3 | 37.7 | 55.1 | 35.0 | 58.9 | 38.6 | 55.7 | 35.9 |
> | **HiTeA (Qwen2.5-VL)** | ZS | **62.3** | **42.2** | **60.7** | **37.0** | **64.1** | **43.2** | **60.2** | **37.3** |
>
> **Notes:**
> - † Method requires training
> - Supervision types: FS (Fully-Supervised), WS (Weakly-Supervised), US (Unsupervised), ZS (Zero-Shot)
> - Best zero-shot results are highlighted in bold
>
> **4.1 QVHighlights Results: SOTA Performance**
>
> We have conducted experiments on the Validation and Test splits, as shown in the table. HiTeA achieves **60.67% mAP\@0.5** on the Test split, yielding two key observations:
>
> - **State-of-the-art Zero-Shot**: HiTeA establishes a new benchmark for strictly training-free approaches, significantly outperforming previous Zero-Shot methods (e.g., surpassing Moment-GPT by +5.5% mAP@0.5).
>
> - **Surpassing Supervised Baselines**: Notably, our training-free approach even exceeds the fully supervised Moment-DETR (54.8%). This confirms that HiTeA generalizes to complex temporal distributions without task-specific optimization.
>
> **4.2 Analysis of Robustness and Multi-instance Grounding**
>
> We believe these results provide strong evidence that HiTeA is robust to multi-instance grounding and complex temporal distributions, which are central challenges in QVHighlights:
>
> - **High-recall proposal generation via Hierarchical Temporal Decomposition (HTD)**
> HTD decomposes the video into multi-scale temporal segments, ensuring that all potentially relevant moments are included in the candidate pool. This is particularly important in QVHighlights, where a single query may correspond to multiple, temporally separated moments.
> - **Robust semantic scoring via VLM-based saliency estimation**
>
>   - The mAP metric in QVHighlights is highly sensitive to the relative ranking of candidate clips.
>
>   - Our VLM-based scoring function $S_{\text{vlm}}$ acts as a semantic saliency predictor, assigning high confidence scores to clips that are semantically aligned with the query.
>
>   - Unlike regression-based methods that primarily focus on boundary refinement, HiTeA focuses on content relevance, enabling it to assign consistently high scores to every occurrence of the queried event.
>
>   - This design naturally aligns with the “one query, multiple moments” setting by treating it as a semantic retrieval problem rather than a single-instance localization problem.

---

> ### Author Response · Authors · 2025-11-21
> **5. For Weakness 5**
>
> We thank the reviewer for this valuable question. We agree that the rationale behind selecting ViT-B/32, DINOv2, and RAFT-Large should be explicitly linked to the hierarchical levels they represent. Our backbone selection is strictly driven by the **feature properties** required to represent each temporal granularity. We selected representative, off-the-shelf models that best instantiate these specific capabilities.
>
> To address this, we have expanded the methodological justification in Section 3 and added a new ablation study comparing alternative backbones in Appendix 4.3.
>
> **5.1  Mapping Feature Properties to Hierarchical Levels**
>
> Instead of generic representations, we map the distinct mechanisms of each model to the physical definition of the hierarchy, leveraging their established capabilities:
>
> - **Event Level $\to$ ViT-B/32**: Events are defined by narrative descriptions (e.g., "cooking pasta"). We utilize the ViT model for its Language-Image Alignment. This ensures that visual representations are directly comparable to the text query, enabling the detection of high-level conceptual boundaries that align with the user's language.
>
> - **Scene Level $\to$ DINO-v2**: Scenes are defined by environmental continuity. We select DINO-v2 for its Discriminative Spatial Layout capability. As a self-supervised vision transformer, DINO-v2 captures global geometry and background statistics. This makes it highly sensitive to camera cuts or location shifts, which are the primary indicators of scene transitions.
>
> - **Action Level $\to$ RAFT**: Actions are defined by atomic physical changes. We employ RAFT to extract Optical Flow Magnitude. This provides a direct measurement of pixel displacement between frames. It ensures that action boundaries are determined by instantaneous kinematic activity rather than static visual appearance.
>
> **5.2 Alternatives and Framework Agnosticism**
>
> To demonstrate that our framework is backbone-agnostic and to justify our specific choices against standard alternatives, we conducted a comprehensive ablation study on the Charades-STA dataset.
>
> We replaced our default models with representative alternatives for each level: FlowNet (vs. RAFT) for the Action level, iBOT (vs. DINOv2) for the Scene level, and Swin-Transformer (vs. ViT) for the Event level.
>
> | Model | Level | 0.3 | 0.5 | 0.7 | mIoU |
> |-------|-------|-----|-----|-----|------|
> | Flownet[1] | Action | 68.92 | 46.94 | 23.41 | 45.02 |
> | iBOT [2] | Scene | 69.14| 46.37| 23.47| 44.96|
> | Swin-Transformer[3] | Event | 67.45 | 45.46 | 23.76 | 44.47 |
> | **Ours** | | **69.62** | **48.52** | **25.59** | **46.29** |
>
> **References**
>
> [1] Dosovitskiy et al. *Flownet: Learning optical flow with convolutional networks.* ICCV 2015.
>
> [2] Zhou et al. *iBOT: Image BERT Pre-training with Online Tokenizer.*
>
> [3] Liu et al. *Swin transformer: Hierarchical vision transformer using shifted windows.* ICCV 2021.
>
> The results show that even with alternative backbones, our method maintains competitive performance (mIoU > 44.4). This empirically verifies that our core contribution lies in the hierarchical processing strategy itself (Event-Scene-Action) rather than being tied to specific feature extractors.
>
> In summary, while the framework is flexible, we selected ViT, DINOv2, and RAFT as they are established, high-performance representatives that maximize the model's overall potential. However, any encoder providing comparable event, scene, action cues could be plugged into the HTD module.

---

> ### Author Response · Authors · 2025-11-21
> **6. For Weakness 6**
>
> We thank the reviewer for raising this critical practical concern. We fully agree that computational efficiency is a prerequisite for real-world deployment. In response, we have added a dedicated "Scalability and Runtime Analysis" subsection in App. A.6 and Tab. 9.
>
> Below, we clarify the pipeline design and provide the requested quantitative evidence to demonstrate that HiTeA remains highly efficient, particularly for long videos.
>
> | Method | Mechanism | Online Complexity | Inference Latency (s / query) | |
> |--------|-----------|------------------|------------------------------|------------------------------|
> | | | | Short (QVH) | Long (TACOS) |
> | **BLIP-VMR** | Sparse 1s scoring + merging | $O(L) \times \text{BLIP}$ | 12.7 | $25$ (est.) |
> | **TFVTG** | Dense scanning | $O(L^2)$ | - | - |
> | **Luo et al.** | Cartesian product | $O(L_v \cdot L_Q)$ | - | - |
> | **Moment-GPT** | Dense frame captioning | $O(L) \times \text{MLLM}$ | $16.1$ | $32$ (est.) |
> | **HiTeA (Ours)** | **Cascade reasoning** | $\mathbf{O(K)}, K{=}9$ | **See below** | **See below** |
> | *--- HiTeA breakdown: offline vs. online ---* | | | | |
> | **1. Offline cost** | ViT + DINO + RAFT | $O(L)$ | $3.0 \text{--} 4.0$ | $6.0 \text{--} 7.0^{\dagger}$ |
> | **2. Online latency** | **VideoCLIP filter $\to$ VLM** | $\mathbf{O(K)}$ | **$7.0 \text{--} 8.0$** | **$7.0 \text{--} 8.0$** |
>
> **Notes:**
> - $^{\dagger}$ Dominant cost is RAFT ($\sim$5--6s). Without flow, offline cost drops to $\sim$1--2s. HTD overhead is negligible ($\sim$0.0012s)
> - For Luo et al., $L_v$ denotes the fixed number of video snippets (32 in their implementation) and $L_Q$ is the number of simple queries
> - The inference latencies on TACOS for BLIP-VMR and Moment-GPT are estimated based on average video length
> - QVHighlights and TACOS have average lengths of $\sim$150s and $\sim$368s, respectively
>
> **6.1 Decomposition of the "Five-Model" Pipeline**
>
> While HiTeA leverages five pre-trained components, they do not operate simultaneously. As clarified in the revision, the workflow is decoupled into two distinct stages:
>
> - **Stage 1: Offline Feature Extraction (One-time cost)**. ViT, DINO, and RAFT are used solely for extracting event, scene, and motion features. These are computed **once per video** and cached. They are not invoked during user interaction and can be parallelized.
>
> - **Stage 2: Online Reasoning (Per-query cost)**. During the query phase, only the lightweight VideoCLIP retriever and the VLM (Qwen2.5-VL) are active. Consequently, the "five-model" architecture does not translate to five-fold latency during actual usage.
>
> **6.2 Quantitative Runtime Analysis (App. A.6)**
>
> We report the specific runtime breakdown on the long-video dataset TACOS (avg. length ≈ 368s):
>
> - **Offline Cost**: Feature extraction takes roughly 6–7s per video (dominated by RAFT). Without optical flow, this drops to 1–2s.
> - **Online Latency**: The user-perceived latency is approximately 7–8s per query.
>   - Our proposed modules (HTD and CR) are extremely lightweight, adding negligible overhead ($\approx$ 1ms and 0.1ms, respectively).
>   - ≈ 99% of the time is spent on the VLM evaluations.
>   - This 7–8s latency is based on standard inference. With optimized libraries or API calls, VLM inference could theoretically drop to ~1s, significantly accelerating the total pipeline.
>
> **6.3 Superior Scalability Compared to Baselines**
>
> A key advantage of HiTeA is that its online complexity relies on a fixed number of candidate evaluations (K), rather than the video length ($L$).
>
> - **Baselines** ($O(L)$ or $O(L^2)$): Methods like Moment-GPT (dense captioning) or TFVTG (dense scanning) scale linearly or quadratically with video length. On QVHighlights, Moment-GPT and BLIP-VMR require **16.7s and 12.7s** per query, respectively.
>
> - **HiTeA** ($O(K)$): By filtering candidates first, HiTeA maintains a stable inference time ($\approx$ 7.5s) regardless of whether the video is **short (QVHighlights)** or **long (TACOS)**.

---

### Official Review · Reviewer_g3Bh · 2025-11-01

**Soundness:** 3
**Presentation:** 3
**Contribution:** 3
**Rating:** 4
**Confidence:** 3

**Summary:**

This paper presents HiTeA, a training-free framework for temporal grounding in long, untrimmed videos. The key innovation is a Hierarchical Temporal Decomposition (HTD) module that structures videos into multi-scale temporal units (events, scenes, actions) using off-the-shelf feature extractors. Candidate segments are scored using frozen vision-language models (VLMs), and a Candidate Refinement module produces final predictions. The method achieves strong results on both long-video benchmarks (Ego4D-NLQ, TACoS) and short-video datasets (Charades-STA, ActivityNet-Captions).

**Strengths:**

1) **Well-motivated problem.** The paper targets a real gap in current approaches—their difficulty scaling to long, temporally complex videos without costly supervised training—and clearly motivates a training-free alternative.

2) **Strong empirical performance.** HiTeA delivers compelling results, especially on long videos: 44.94% R@0.1 on TACoS (a +12.4% absolute gain).

3) **Robust generalization.** The method transfers across diverse domains and durations—from short clips (≈29 s, Charades-STA) to extended egocentric footage (≈472 s, Ego4D-NLQ)—indicating strong robustness to video type and length.

**Weaknesses:**

1) **Limited technical novelty.**
   While the system is effective, its building blocks are largely established:
   - Hierarchical temporal decomposition has prior art.
   - Zero-shot use of pretrained VLMs is well explored.
   - Change-point detection (e.g., PELT) is off-the-shelf.
   - The overall pipeline follows a familiar proposal-generation + scoring pattern.

2) **Method complexity and engineering burden.**
   The framework stitches together multiple components (three feature extractors, VideoCLIP filtering, VLM scoring, candidate refinement) and exposes many hyperparameters (α, β, λ, k, thresholds). This raises concerns about:
   - **Generalizability** beyond the tested datasets,
   - **Sensitivity** to hyperparameters (only β and λ receive analysis), and
   - Whether the gains stem from **principled design** versus careful tuning and component stacking.

3) **Unclear rationale for hierarchy effects across lengths.**
   Table 7 shows hierarchical merging helps long videos but harms short ones; the discussion in A.4.2 reads as a post-hoc explanation rather than a principled, testable design hypothesis.

4) **Domain specificity of the three-level hierarchy.**
   The event–scene–action granularity—especially the scene level—appears tailored to the evaluated domains. It’s unclear how this hierarchy transfers to other video genres (e.g., sports broadcasts, surveillance, instructional videos) without re-defining levels or retuning cues.

**Questions:**

Please refer to the Weaknesses.

---

> ### Author Response · Authors · 2025-11-21
> **Summarize**
>
> We sincerely thank the reviewer for the constructive comments and the encouraging evaluation. We are encouraged by your recognition of HiTeA’s **“strong empirical performance”** and its **“robust generalization”** across diverse domains and lengths. Regarding the concerns raised, we provide the following clarifications:
>
> - **Response to W1 (Novelty)**: We clarify that our contribution is not in inventing the individual components (VLMs/PELT), but in **orchestrating** them into a unique, **training-free algorithmic scaffold** that effectively solves the long-video scalability problem—something prior methods failed to achieve.
>
> - **Response to W2 (Complexity & Sensitivity)**: We emphasize that HiTeA uses **one fixed hyperparameter configuration** across all 5 diverse benchmarks, demonstrating that the gains stem from a **principled architectural design** rather than fragile tuning or component stacking.
>
> - **Response to W3 (HM Rationale)**: We explain that the length-adaptive HM strategy is **a design choice** derived from the distinct structural properties of long vs. short videos, a hypothesis that is consistently validated by our empirical results.
>
> - **Response to W4 (Domain Generality)**: We highlight that our hierarchy is generic and that our evaluation **already covers the reviewer's suggested genres** (Sports in ActivityNet, Instructional in TACoS/Ego4D), proving the framework's broader applicability.

---

> ### Author Response · Authors · 2025-11-21
> **1. For Weakness 1**
>
> We thank the reviewer for pointing out that several components in HiTeA have strong prior art. We acknowledge that HiTeA leverages established pre-trained backbones. However, our contribution is instead at the **system and algorithmic level**: we show how to orchestrate these established tools into a training-free architecture that (i) scales to long videos and (ii) for the first time lets a fully training-free system surpass supervised methods on benchmarks.
>
> Below, we clarify how HiTeA differs from existing uses of each building block.
>
> **1.1 "Hierarchical temporal decomposition has prior art."**
>
> While hierarchical modeling is an established concept, prior VTG methods typically learn **implicit** hierarchies within end-to-end networks. This paradigm relies heavily on **dense supervision (training on thousands of annotated clips)** and is tightly coupled to specific backbones.
>
> **In contrast, HiTeA introduces a fully training-free algorithm.** Instead of learning a hierarchy from data, we explicitly extract distinct visual cues for  specific temporal levels (Event, Scene, Action). These levels are aligned via a top-down merging operator to enforce strict containment. This design proves that explicitly engineered signals can surpass learned features in zero-shot settings.
> Thus, while the concept of hierarchy is prior art, our specific **multi-cue orchestration** is novel and essential for achieving high accuracy without training.
>
> **1.2 "Zero-shot use of pre-trained VLMs is well explored."**
>
> Recent zero-shot VTG methods either (i) train small heads with pseudo-labels, or (ii) directly query pre-trained CLIP-style models or MLLMs on uniformly/loosely sampled clips. These approaches typically assume **short clips** (Charades-STA, ANet) and uniform sampling or **run dense captioning / QA with an MLLM** on many frames (e.g., Moment-GPT).
>
> **HiTeA fundamentally alters this paradigm by decoupling perception and reasoning.**
>
> - Heavy perception (feature extraction) is performed once offline.
>
> - Online reasoning is sparse: the VLM reasons only over a fixed number of high-quality HTD candidates.
>
> This structural innovation reduces online complexity to $O(K)$ (independent of video length). It allows a fully training-free system to achieve **20.12 R@1** on Ego4D-NLQ and **44.94 R@1** on TACoS, surpassing both strong supervised baselines and previous zero-shot methods that rely on expensive dense scanning.
>
> **1.3 “Change-point detection (e.g., PELT) is off-the-shelf.”**
>
> We agree that PELT itself is standard. Our contribution is how this tool is used inside a broader temporal reasoning scheme.
>
> We apply PELT **only at specific levels** (scene & action) and combine it with a different detector at the event level, reflecting the different temporal statistics of scenes vs. actions vs. events. These boundaries are then passed through the **hierarchical merging module**, turning three noisy change-point sets into a coherent, multi-granularity scaffold.
>
> So PELT is indeed off-the-shelf, but its **role inside HTD is not**: it is part of a carefully engineered, multi-signal decomposition pipeline that directly underpins both the accuracy gains (Tab. 4 in revised paper) and the candidate-reduction ratios in Tab. 5.
>
> **1.4 “The overall pipeline follows a familiar proposal-generation + scoring pattern.”**
>
> We acknowledge that HiTeA fits the high-level "generate-then-score" abstraction. However, our innovation lies in the **nature of proposals** and **the interaction between scoring and hierarchy**.
>
> - **Structured Proposals (vs. Uniform Windows)**
>
> Standard methods typically use uniform sliding windows or single-scale heads. In contrast, HiTeA’s proposals are structurally coherent units derived from the multi-granularity HTD scaffold. This fundamental difference yields candidates that are (a) **significantly fewer**, (b) structurally consistent, and (c) length-adaptive. This is critical for long videos where dense sliding windows generate prohibitive redundancy.
>
> - **Progressive, Semantic Aggregation (vs. Independent Scoring)**
>
> Unlike prior work that scores proposals independently (often followed by NMS), HiTeA employs a hierarchy-aware mechanism: (a) A lightweight VideoCLIP pre-filter scores HTD units and merges neighbors, reducing candidates by 76.6% (38.2 $\rightarrow$ 8.9 on TACoS) with negligible cost. (b) The VLM then reasons over the top candidates using Candidate Refinement. Crucially, CR progressively re-aggregates units if they are semantically consistent. This **addresses the limitation of structural boundaries**, as natural-language events often span multiple structural units.
>
> - **Quantitative Impact**
>
> This logic is not merely cosmetic. As shown in Tab. 4, adding the semantic CR module atop the structural HTD yields **+2.8%** mIoU on TACoS and **+4.7%** mIoU on Charades-STA. This confirms that our "scoring" is a necessary semantic merging process that complements the structural decomposition.

---

> ### Author Response · Authors · 2025-11-21
> **2. For Weakness 2**
>
> We appreciate the reviewer’s scrutiny regarding system complexity. We wish to clarify that the architecture is driven by **robustness and scalability**, not parameter engineering. We address the concerns below.
>
> **2.1 Generalizability beyond the tested datasets**
>
> HiTeA exhibits exceptional stability without fragile per-dataset tuning. Instead, we adopt a simple two-regime configuration based solely on video duration:
>
> - **For Short Videos**: One fixed setting is applied to Charades-STA, ActivityNet-Captions, and QVHighlights.
>
> - **For Long Videos**: A separate fixed setting is applied to TACoS and Ego4D-NLQ to account for the order-of-magnitude difference in temporal scope.
>
> We do not re-tune values for specific datasets or splits. This demonstrates that HiTeA relies on length-adaptive logic rather than dataset-specific hacking, proving that the architecture generalizes robustly as long as the broad temporal scale is known.
>
> **2.2 Sensitivity to hyperparameters**
>
> Most hyperparameters are either set by simple, interpretable rules, or empirically shown to be insensitive within a broad range.
>
> - **Top-k candidates (K)**: We choose K=9, but it is not finely tuned per dataset. Actually, varying K in a reasonable range (e.g., 5, 9, 13) yields only minor changes in performance.
>
> - **Temporal tolerance α**: This controls the soft boundary matching. We set it once based on typical segment durations and do not modify it across benchmarks, as long as it is within a reasonable scale of the average segment length.
>
> - **β and λ**: We focus our detailed sensitivity study on the hyperparameters β and λ. As illustrated in Fig. 4, the method is **robust** to these variations, with IoU scores exhibiting **minimal fluctuation** across the tested ranges.
>
> Overall, HiTeA demonstrates strong robustness to hyperparameter variations, proving that no narrow "sweet spot" is required to sustain the reported performance.
>
> **2.3 Principled design vs. “component stacking”**
>
> We acknowledge that HiTeA integrates multiple components. However, this design is driven by the **necessity** to decouple distinct temporal granularities in long videos.
>
> - **Functional Specialization of Features**
>
> Video understanding requires capturing Action, Scene, Event, which operate at different timescales. No single off-the-shelf model currently captures all three effectively, we therefore employ a triad of specialized models: RAFT for fine-grained Action, DINO for contextual Scene, and ViT for high-level Event representation.
>
> Ablation studies (Tab. 4 in revised paper) demonstrate that these components are **non-redundant**: removing any single level leads to a significant drop in performance. Thus, the multi-component design is the minimal requirement to cover the full temporal spectrum of long videos.
>
> - **Efficiency through Decomposition**
>
> Paradoxically, using multiple specialized components improves overall efficiency. By using lightweight detectors (HTD) to filter the search space, we avoid asking the computationally expensive VLM to process the entire video.
>
> This transforms the task from a dense scan (scaling with $O(L)$ or $O(L^2)$) to a sparse selection ($O(K)$), effectively reducing inference latency on TACoS to 7–8s (vs. ~30s for baselines). This strategy strategically shifts the computational burden to **offline** feature extraction, ensuring that **online** inference remains fast and scalable.
>
> Together,  these architectural decisions allow HiTeA to scale to long videos with **fixed inference overhead**, demonstrating that the system's efficacy stems from a **cohesive structural design** rather than a simple aggregation of modules.

---

> ### Author Response · Authors · 2025-11-21
> **3. For Weakness 3**
>
> We thank the reviewer for the comments. The performance divergence is not a post-hoc explanation but stems from the **inductive bias** of Hierarchical Merging (HM). HM imposes a structured prior (Events ⊃ Scenes ⊃ Actions) that aligns with long videos but mismatches the flat temporal nature of short videos.
>
> **3.1 Design Intent: Matching Inductive Bias to Data Granularity**
>
> HM is explicitly **designed to handle the complexity of long, untrimmed videos**. These videos possess a deep temporal hierarchy where organizing candidates into a structure reduces redundancy and captures long-range dependencies. Short videos, however, typically consist of a single event or highly overlapping actions (flat structure). Applying a hierarchical constraint to non-hierarchical data is theoretically suboptimal; thus, the method was designed with the expectation that HM is essential for long videos but potentially redundant for short ones.
>
> **3.2 Mechanism: Structural Degeneration in Short Videos**
>
> Disabling HM for short videos is a principled choice driven by **structural degeneration**. Short videos typically consist of a single continuous scene, causing the hierarchical distinction between "events" and "scenes" to effectively vanish (i.e., the event level degenerates into the scene level). Therefore, the multi-level constraints of HM become theoretically redundant rather than essential.
>
> **It is important to note that our method remains highly robust: even with HM enabled, it achieves strong performance that outperforms most prior arts.** However, disabling HM allows the model to bypass these unnecessary constraints on an already concise structure, preventing the strictly hierarchical merging of valid candidates and yielding a further performance gain.
>
> **3.3 Empirical Validation vs. Post-hoc Explanation**
>
> Our experimental results validate this pre-design hypothesis rather than contradict it. We observe clear gains on long-video datasets (where the hierarchy exists) and neutral/slight drops on short-video datasets (where hierarchy is absent) when HM is active. This consistent behavior confirms that HM functions exactly as intended: it imposes order where needed (long videos) but should be bypassed where granularity is paramount (short videos). We revised the App. A.4.2 to explicitly articulate this design rationale.

---

> ### Author Response · Authors · 2025-11-21
> **4. For Weakness 4**
>
> We appreciate the reviewer highlighting specific domains like sports, instructional videos, and surveillance. We clarify below that our current evaluation benchmarks already cover the majority of these genres, and our hierarchical design is generic.
>
> **4.1 Substantial Coverage of Suggested Genres in Current Benchmarks**
>
> While named "ActivityNet" or "TACoS," the datasets evaluated in the paper directly correspond to the genres in question, validating the method's transferability:
>
> - **Regarding Sports Broadcasts**: ActivityNet-Captions explicitly contains numerous sports categories (e.g., long jump, playing basketball, gymnastics, discus throw). Many of these clips are sourced from broadcast highlights, featuring camera cuts and distinct event boundaries.
>
> - **Regarding Instructional Videos**: TACoS and Ego4D are inherently instructional. TACoS consists entirely of cooking procedures (step-by-step instructions), while Ego4D covers egocentric "how-to" scenarios (e.g., bike repair, DIY tasks).
>
> **4.2 Generality of the Event–Scene–Action Hierarchy**
>
> The hierarchy is not domain-specific but rather a universal temporal scaffold:
>
> - **Events** capture high-level activities (applicable to sports matches or instructional goals).
>
> - **Scenes** capture contextual/camera changes (crucial for sports cuts or instructional step changes).
>
> - **Actions** capture atomic movements (ubiquitous in surveillance).
>
> Since HiTeA uses general-purpose feature extractors without dataset-specific tuning, the framework is conceptually ready to be applied to pure surveillance data without architectural changes.
>
> **4.3 Robustness to non-strict hierarchies via HTD and CR**
>
> We agree that real-world videos need not obey a strict nested structure (Event ⊃ Scene ⊃ Action). HiTeA is designed with this in mind:
>
> - **Hierarchical Temporal Decomposition (HTD)** provides an initial multi-scale segmentation into event-/scene-/action-like units based on temporal structure, without assuming perfect containment.
>
> - **Candidate Refinement (CR)** then adapts these segments to the query, allowing merges and adjustments when, for example, a single event spans multiple scenes (frequent in sports broadcasts with many camera cuts) or when multiple actions overlap or interleave (common in surveillance-like settings).
>
> Thus, the hierarchy plays the role of a flexible multi-scale prior rather than a rigid ontology, which is critical for transferring to less-structured domains.
>
> **4.4 Extensibility to other genres (including surveillance)**
>
> Finally, the framework is modular: **domain-specific models** (e.g., models specialized for surveillance or professional sports broadcasting) can be plugged into the same Event–Scene–Action scaffold without altering the core algorithm. This makes it straightforward to extend HiTeA to pure surveillance or other specialized genres in future work.

---

### Official Review · Reviewer_wbqj · 2025-11-03

**Soundness:** 3
**Presentation:** 3
**Contribution:** 3
**Rating:** 6
**Confidence:** 4

**Summary:**

This paper introduces HiTeA, a training-free framework for long-video temporal grounding. The method tackles the challenge of locating temporal segments in long, untrimmed videos corresponding to natural language queries—without requiring any supervised training.

**Strengths:**

The Hierarchical Temporal Decomposition (HTD) effectively aligns video segments with textual queries at multiple granularities, bridging the gap between semantic understanding and temporal localization.

**Weaknesses:**

1. **Feature extraction choices** – Why were **ViT**, **DINO**, and **FLOW** specifically chosen for extracting event-level, scene-level, and action-level features, respectively? How do these choices contribute to the complementary strengths of each temporal layer?
2. **Inference efficiency** – The paper discusses computational complexity analytically, but could you provide a **runtime comparison** with representative baselines on long-video datasets to empirically demonstrate HiTeA’s scalability?

**Questions:**

Please refer to the points listed under *Weaknesses*.

---

> ### Author Response · Authors · 2025-11-21
> **Summarize**
>
> We sincerely thank the reviewer for the constructive feedback and the positive evaluation. We are deeply encouraged by your recognition of our **Hierarchical Temporal Decomposition (HTD) design** for effectively "bridging the gap between semantic understanding and temporal localization." Your questions regarding feature rationale and efficiency have guided us to strengthen the theoretical and practical justifications of our framework:
>
> - **Regarding feature choices (W1)**: We selected these models to enforce a principled **Modality-Granularity Alignment**, explicitly mapping RAFT, DINO, and ViT to the Action, Scene, and Event levels, respectively. This combination ensures complementary coverage—capturing **how** (action), **where (scene)**, and **what (event)**—while remaining agnostic to specific backbone architectures.
>
> - **Regarding inference efficiency (W2)**: We have added a **runtime benchmark on the long-video dataset TACOS** (App. A.6). The results demonstrate that HiTeA achieves a **stable, length-insensitive online latency (~7-8s)**, significantly outperforming representative baselines, which suffer from linear or quadratic complexity scaling.

---

> ### Author Response · Authors · 2025-11-21
> **1. For Weakness 1**
>
> We thank the reviewer for this opportunity to clarify our design rationale. The selection of RAFT, DINO, and ViT-CLIP is not arbitrary; it is driven by the need to capture three fundamental and complementary visual cues that naturally align with our temporal hierarchy.
>
> **1.1 Mapping features to temporal granularity**
>
> We clarify this mapping more explicitly in the revised App. A.3.1:
>
> - **Action Level (via RAFT)**:
>
> Actions are primarily defined by short-term physical movement (e.g., “opening a door”). We choose RAFT because it provides dense, pixel-level optical flow, making it highly sensitive to fine-grained motion boundaries that appearance/semantic models often smooth out.
>
> - **Scene Level (via DINO)**:
>
> Scenes are defined by **visual context and appearance consistency** (e.g., staying within the same room or viewpoint). We choose DINO because its self-supervised features are exceptionally robust at capturing global visual coherence and salient object identity. This makes it ideal for identifying boundaries where the visual environment shifts (e.g., shot cuts or spatial transitions), serving as the structural "container" for actions.
>
> - **Event Level (via ViT)**:
>
> Events represent high-level, language-driven concepts that often abstract over varying visual appearances (e.g., "preparing a meal" spans chopping, cooking, and plating). We use ViT for its vision-language alignment. **By continuously computing the similarity between the current frame and the preceding segment's prototype**, we can track the evolution of the narrative. This allows HiTeA to ignore local visual changes (handled by DINO) and place boundaries only when the high-level semantic topic shifts.
>
> These three features form an orthogonal triad—Event (What), Scene (Where), and Action (How). They provide complementary information: the Action level captures fine-grained motions that are often smoothed over by semantic models, while the Scene level provides a consistent visual context, and the Event level tracks the progression of high-level semantics.Moreover, our framework is architecture-agnostic. The models used are strong, off-the-shelf representatives and can be replaced (e.g., Swin for ViT) without altering HiTeA's core logic.
>
> **1.2 How they form temporal complementarity**
>
> These choices lead to complementary strengths across temporal scales:
>
> - The action level (RAFT) focuses on **fine, short-term motion**.
>
> - The scene level (DINO) tracks mid-level visual **contextual** changes.
>
> - The event level (ViT-CLIP) captures long-range **semantic** evolution.
>
> As shown in our ablation in Tab. 4(revised paper), each single level (event / scene / action) already improves mIoU over the baseline without HTD/CR. Combining two levels yields further gains, and using **all three levels** together achieves the **best** performance. For example, compared with the baseline, using the full three-level HTD with CR improves mIoU from **14.87 to 19.79** on TACoS and from **35.07 to 46.29** on Charades-STA. This demonstrates that no single feature dominates; instead, they provide complementary cues across distinct levels of video understanding. This complementary design—where RAFT, DINO, and ViT capture action, scene, and event levels respectively—is key to the robustness and performance of HiTeA, and we have added this explanation to App. A.3.1.

---

> > ### Comment · Reviewer_wbqj · 2025-11-27
> >
> > Thank you for the detailed clarification in your response. I appreciate the additional explanation and the updates in the revised appendix.
> >
> > Regarding Weakness 1, while the overall motivation of the hierarchical design is clear, the specific choice of ViT / DINO / RAFT (or FLOW) as representatives of the three temporal granularities is still not fully convincing to me. The current justification feels more post-hoc than principled. Since this feature triad is central to your method, I believe a more explicit mechanism that directly reflects the motivation—or at least an ablation comparing different model choices at each level—would make the argument significantly stronger.
> >
> > For Weakness 2, I agree that the new runtime numbers behave as expected, but the explanation in the original submission was somewhat brief, and the revised submission has already exceeded the 9-page limit. Because these clarifications were not fully integrated into the main paper and some points remain insufficiently justified, I decided to slightly lower the score.
> >
> > I still think the paper has interesting ideas, and I appreciate the efforts you made in the rebuttal.

---

> ### Author Response · Authors · 2025-11-21
> **2. For Weakness 2**
>
> We appreciate the reviewer’s practical focus on scalability. In response, we have included a dedicated runtime study on the long-video dataset TACOS (avg. 368s) in App. A.6 and Tab. 9. The results demonstrate that HiTeA achieves superior efficiency compared to baselines.
>
> **2.1 Empirical runtime on long videos**
>
> We now report a detailed runtime breakdown for HiTeA on TACOS, as shown in the table:
> | Method | Mechanism | Online Complexity | Inference Latency (s / query) | |
> |--------|-----------|------------------|------------------------------|------------------------------|
> | | | | Short (QVH) | Long (TACOS) |
> | **BLIP-VMR** | Sparse 1s scoring + merging | $O(L) \times \text{BLIP}$ | 12.7 | $25$ (est.) |
> | **TFVTG** | Dense scanning | $O(L^2)$ | - | - |
> | **Luo et al.** | Cartesian product | $O(L_v \cdot L_Q)$ | - | - |
> | **Moment-GPT** | Dense frame captioning | $O(L) \times \text{MLLM}$ | $16.1$ | $32$ (est.) |
> | **HiTeA (Ours)** | **Cascade reasoning** | $\mathbf{O(K)}, K{=}9$ | **See below** | **See below** |
> | *--- HiTeA breakdown: offline vs. online ---* | | | | |
> | **1. Offline cost** | ViT + DINO + RAFT | $O(L)$ | $3.0 \text{--} 4.0$ | $6.0 \text{--} 7.0^{\dagger}$ |
> | **2. Online latency** | **VideoCLIP filter $\to$ VLM** | $\mathbf{O(K)}$ | **$7.0 \text{--} 8.0$** | **$7.0 \text{--} 8.0$** |
>
> **Notes:**
> - $^{\dagger}$ Dominant cost is RAFT ($\sim$5--6s). Without flow, offline cost drops to $\sim$1--2s. HTD overhead is negligible ($\sim$0.0012s)
> - For Luo et al., $L_v$ denotes the fixed number of video snippets (32 in their implementation) and $L_Q$ is the number of simple queries
> - The inference latencies on TACOS for BLIP-VMR and Moment-GPT are estimated based on average video length
> - QVHighlights and TACOS have average lengths of $\sim$150s and $\sim$368s, respectively
>
> **2.2 HiTeA runtime decomposition on long videos**
>
> HiTeA maintains a low and stable latency even for long videos. As detailed in the table, the runtime consists of two parts:
>
> **Offline Feature Extraction (~6-7s)**: This is a one-time cost per video (dominated by RAFT) and is fully parallelizable.
>
> **Online Query Latency (~7-8s)**: This is the actual user-perceived latency. It is dominated by the Qwen2.5-VL inference, while our proposed HTD and CR algorithms introduce negligible overhead ($\sim 10^{-3}$s). Crucially, because HiTeA processes a fixed number of candidate events rather than scanning every frame, this cost is **insensitive to video length**, remaining stable at 7-8s whether the video is short or long.
>
> **2.3 Comparison with baselines**
>
> In contrast, representative baselines exhibit significant scaling issues on long videos like TACOS:
>
> - **Dense Captioning Methods (BLIP-VMR, Moment-GPT)**: These methods follow an $O(L)$ complexity, processing videos linearly. Based on their processing speeds, they require approx. 25s-32s per query on TACoS slower than HiTeA.
>
> - **Dense Proposal Methods (TFVTG, Luo et al.)**: These methods generally scale quadratically $O(L^2)$ or rely on dense scanning. Their computational cost becomes prohibitive for long videos (marked as “-” in table as these runtimes are unavailable in official reports).

---

> ### Author Response · Authors · 2025-11-27
>
> We sincerely appreciate your continued engagement and your acknowledgment of the efforts made in our initial rebuttal. We fully appreciate your perspective regarding the principled justification of our model choices (ViT/DINO/RAFT) and the integration of the runtime analysis.
>
> We plan to address these two core issues in the new version as follows:
>
> **1. On Model Selection Principles and Ablation Studies (Weakness 1)**
>
> Our core contribution lies in the hierarchical processing strategy itself (Event-Scene-Action). While other models could theoretically serve similar functions within this framework, we selected ViT, DINO, and RAFT because they are established representatives with proven generalization capabilities in their respective domains. This choice ensures a robust implementation that effectively validates our hierarchical design.
>
> To empirically support this, we will include a comparative ablation study in the final revision (e.g., RAFT vs. FlowNet for the Action level). We expect this analysis to demonstrate that while the hierarchical structure is the primary driver of success, leveraging these specific, high-performance representatives is also important for maximizing the model's overall potential, thereby justifying our architectural decisions.
>
> **2. On Runtime Analysis and Page Limits (Weakness 2)**
>
> According to ICLR2026 guidelines, the camera-ready version allows for 10 pages. We will move the detailed runtime efficiency analysis from the appendix directly into the main text. This ensures that all key arguments and justifications are fully presented in the main text without over-condensing the content.
>
> Thank you again for your constructive feedback, which has been instrumental in refining our work.

---

> > ### Comment · Reviewer_wbqj · 2025-11-27
> >
> > Thank you for the clarification.
> >
> > For Question 1, the paper would be more convincing with additional ablation studies.
> >
> > For Question 2, I agree with the authors’ explanation.
> >
> > Overall, I will maintain my original score.

---

> ### Author Response · Authors · 2025-12-02
> **Additional Ablation and Runtime Updates**
>
> We thank the reviewer for the prompt and constructive feedback. We have incorporated suggestions into the final revision as follows:
>
> **1. Response to Question 1**
>
> To empirically justify our model selection and separate the contribution of the architecture from the specific backbones, we conducted an ablation study on Charades-STA. We replaced our default models with standard alternatives: FlowNet (vs. RAFT) for the Action level, iBOT (vs. DINOv2) for the Scene level, and Swin-Transformer (vs. ViT) for the Event level.
>
> | Model | Level | 0.3 | 0.5 | 0.7 | mIoU |
> |-------|-------|-----|-----|-----|------|
> | Flownet[1] | Action | 68.92 | 46.94 | 23.41 | 45.02 |
> | iBOT [2]	| Scene	| 69.14| 	46.37| 	23.47| 	44.96|
> | Swin-Transformer[3] | Event | 67.45 | 45.46 | 23.76 | 44.47 |
> | **Ours** | | **69.62** | **48.52** | **25.59** | **46.29** |
>
> **References**
> [1] Dosovitskiy et al. *Flownet: Learning optical flow with convolutional networks.* ICCV 2015.
>
> [2] Zhou et al. *iBOT: Image BERT Pre-training with Online Tokenizer.*
>
> [3] Liu et al. *Swin transformer: Hierarchical vision transformer using shifted windows.* ICCV 2021.
>
>
> The results show that even with alternative backbones, the method achieves competitive performance (mIoU > 44.4). This robustness empirically verifies that our core contribution lies in the hierarchical processing strategy itself (Event-Scene-Action). While other models can theoretically serve similar functions within this framework, we selected ViT, DINO, and RAFT because they are established representatives with proven generalization capabilities. As shown in the table, leveraging these specific, high-performance representatives maximizes the model's overall potential (achieving 46.29 mIoU), thereby justifying our architectural decisions.
>
> We have included a detailed analysis of these results in Appendix 4.3.
>
> **2. Response to Question 2**
>
> We have moved the summary of the runtime comparison  to Section 4.5 of the main text. Due to the strict page limits, the detailed breakdown of offline vs. online latency and the full comparison table with all baselines are preserved in Appendix A.6.
>
> We believe these revisions fully address the remaining concerns and strengthen the paper's contributions.

---

### Official Review · Reviewer_KVaa · 2025-11-04

**Soundness:** 3
**Presentation:** 3
**Contribution:** 3
**Rating:** 8
**Confidence:** 3

**Summary:**

This paper proposes HiTeA, a training-free framework for Video Temporal Grounding (VTG) in long videos. The authors argue that existing methods either rely on expensive supervised annotations or perform poorly when handling the complex temporal structures and content redundancy in long videos. The core contribution of HiTeA is a Hierarchical Temporal Decomposition (HTD) mechanism. This mechanism simulates human search strategies, decomposing the video into three granularities: 'events', 'scenes', and 'actions'. It first uses various pre-trained models (like ViT, DINO, RAFT) to extract multi-level features and generate multi-scale candidate segments. Subsequently, the framework adopts a two-stage strategy: first, it pre-filters candidates using a lightweight VideoCLIP, and then it scores the candidates using a frozen large vision-language model (VLM). Experiments show that HiTeA significantly outperforms existing zero-shot (ZS) methods on multiple short- and long-video benchmarks. For example, on the TACOS dataset, it achieves a 12.4% absolute performance gain. More importantly, on long-video benchmarks like Ego4D-NLQ, the method achieves performance competitive with or even superior to state-of-the-art (SOTA) supervised methods.

**Strengths:**

-  **Significance and Impact:** The paper's strongest merit lies in its **groundbreaking results**. As a **training-free** framework, HiTeA's mIoU (8.12%) on Ego4D-NLQ (a highly challenging long-video benchmark) **surpasses** all listed fully-supervised SOTA methods (e.g., UniVTG's 4.91%). This challenges the traditional notion that SOTA performance must rely on dense supervision and holds significant implications for the VTG field.
- **Originality and Intuitive Method:** The core **HTD (Hierarchical Temporal Decomposition)** idea is highly novel. It does not blindly segment the video; instead, it emulates the human search intuition of progressing from 'event' to 'scene' to 'action' and elegantly maps this intuition to different pre-trained features (ViT for semantics, DINO for structure, RAFT for motion). This is a very elegant design.
- **High-Quality and Rigorous Experiments:**
    * **Main experiments** convincingly demonstrate its SOTA performance on both long and short videos.
    * The **ablation study (Table 3)** is solid. It clearly quantifies the contribution of each layer in HTD (Event, Scene, Action) and the necessity of the CR module, proving the effectiveness of each component in the framework.
    * The **efficiency analysis (Table 4)** adds significant practical value, showing that VideoCLIP pre-filtering reduces the computational load on Ego4D by 93.9%, addressing the pain point of slow VLM inference on long videos.

-  **Clarity:** As mentioned, the paper is well-written and logically organized. **Figures 1 and 2**, in particular, intuitively illustrate the difference between HiTeA and traditional methods, as well as its complex internal workflow.

**Weaknesses:**

My criticisms are  primarily focus on the rigor of its claims and the transparency of its details.

-   **Contradiction between the motivation for Score Fusion (Sec 3.5) and the results (Sec 4.4):**
    * **Issue:** In Sec 3.5, the authors introduce $s_{final}=\lambda \cdot s_{vlm}+(1-\lambda) \cdot s_{clip}$, motivated by the idea of fusing the VLM's semantic information with VideoCLIP's structural information.
    * **Contradiction:** However, in the parameter analysis in Sec 4.4 (Fig 4, Right), the optimal value for $\lambda$ is set to **0.99**. This implies that the contribution of $s_{clip}$ is almost negligible (1%).
    * **Suggestion:** The authors also admit in the appendix that $s_{clip}$ mainly serves as a "tie-breaker". It is suggested that the authors state this more candidly in the main text (Sec 3.5)—i.e., that the VLM is absolutely dominant and CLIP is only used for filtering and tie-breaking—rather than presenting it as a "balanced fusion." This would make the paper's motivation more rigorous.

-   **A key implementation detail is hidden in the appendix:**
    * **Issue:** Appendix A.4.2 (Table 7) reveals that for short videos (Charades-STA, ANet), the **Hierarchical Merging (HM) module is disabled** because it slightly harms performance.
    * **Suggestion:** This is a crucial implementation detail that reveals the framework's adaptability (prioritizing structure for long videos, diversity for short videos). This should not be hidden in the appendix. It is suggested that the authors move this detail (i.e., that HM is disabled for short videos) to the main paper's experimental setup (Sec 4.1) or method (Sec 3.3) for discussion.

**Questions:**

My main concerns are articulated in the "Weaknesses" section and are more about presentation and rigor. Here they are converted into specific questions for the authors' rebuttal:

-   **Regarding $\lambda=0.99$:** Given the optimal value of $\lambda$ is 0.99, this suggests the contribution of $s_{clip}$ is minimal. Does this imply that the "fusion" motivation proposed in Sec 3.5 is not practically realized, and the true role of $s_{clip}$ (as mentioned in the appendix) is merely for filtering and tie-breaking? Please clarify the exact role of $s_{clip}$ in the final score.
-  **Regarding the disabling of HM:** You mention in Appendix A.4.2 that the HM module is disabled for short videos because preserving the diversity of independent features is more effective than enforcing a hierarchy. Does this mean the best practice for HiTeA involves using **two different configurations based on video length (long/short)**? Could you more explicitly state this (very reasonable) design choice in the main text?
-  **Regarding the complexity of CR and HTD:** Your framework appears to have two "merging" steps: (1) the hierarchical merging in HTD (Alg 1) and (2) the progressive merging in CR (Alg 2). This makes the method (especially the CR module) seem somewhat complex. Table 3 shows that CR is crucial, but could you provide further intuition on why the merging in HTD is insufficient, necessitating a second round of merging in CR?

---

> ### Author Response · Authors · 2025-11-21
> **Summarize**
>
> We sincerely thank the reviewer for the insightful comments and the positive evaluation. We are encouraged by your recognition of HiTeA’s **“groundbreaking results”** as a training-free framework and the **“originality”** of our Hierarchical Temporal Decomposition (HTD) design. We also appreciate the positive remarks regarding the rigor of our experiments and the clarity of the presentation.
>
> And your constructive comments have helped us clarify our design choices and improve the paper's transparency.
> - **Regarding the fusion weight (W1/Q1)**: We clear up that the method follows a **VLM-dominant design**, where the VideoCLIP term acts as a crucial **tie-breaker** for refining candidate ordering rather than a balanced partner.
> - **For the Hierarchical Merging strategy (W2/Q2)**: We have **moved the discussion to the main text** to explicitly define our **length-adaptive** configuration (enabled for long videos, disabled for short ones).
> - **Finally, regarding the dual merging steps (Q3)**: We clarify their **complementary roles**, distinguishing HTD’s offline video-structure mining from CR’s online query-centric aggregation.

---

> ### Author Response · Authors · 2025-11-21
> **1. For Weakness 1 and Question 1**
>
> We are grateful for this keen point, which helped us better articulate the underlying logic of our design choice.
> **Although the VideoCLIP term is weighted by only 0.01, it has a clear and intentional functional role as a structural tie-breaker on top of a VLM-dominant score.** In our response, we detail (1) the overall design, (2) the interpretation of λ = 0.99, and (3) how we have revised the paper accordingly.
>
> **1.1 Overall design: VLM-dominant with auxiliary structural refinement**
>
> We agree that our original wording “balanced fusion” was misleading given the optimal fusion weight. The method is not designed as a symmetric fusion of two equally important streams. Instead, it follows a VLM-dominant scoring scheme, where:
> - the **VLM score** provides the primary semantic decision, and
> - the **VideoCLIP score** serves as an auxiliary structural signal that refines the ordering among candidates.
>
> In essence, the VLM handles the core semantic judgment of **‘what is correct,’** while VideoCLIP's role is secondary—it merely breaks ties or handles cases of VLM uncertainty.
>
> **1.2 Interpretation of λ = 0.99 and why 1% still matters.**
>
> As shown in our parameter analysis (Sec. 4.4, Fig. 4 right), the optimal performance occurs at λ = 0.99, meaning that the VLM contributes 99% of the final score while VideoCLIP contributes 1%. However, since the VLM consistently dominates, the performance variation between 0.9 and 0.99 is minimal. This is because:
>
> - Qwen2.5-VL provides strong **semantic** judgments but tends to produce **discretized confidences**, often assigning the same high score to multiple candidates.
> - The VideoCLIP score, in contrast, is **continuous** and sensitive to **fine-grained structural** differences between video segments.
>
> Under these conditions, even a weight of 0.01 can meaningfully influence the ranking whenever candidates share similar or identical VLM scores: the continuous VideoCLIP signal slightly perturbs the combined score so that segments with better structural alignment are ranked higher.
>
> **1.3 Revisions made in the paper**
>
> In the revised Sec. 3.5, we now explicitly describe the scoring scheme as "VLM-dominant," with VideoCLIP acting strictly as a structural filter and tie-breaker, and have removed the misleading phrase "balanced fusion." Consistent with this update, Sec. 4.4 now clarifies that the optimal λ = 0.99 quantitatively reflects this design, confirming that the 1% VideoCLIP weight is primarily responsible for resolving ranking ambiguities among candidates with similar VLM confidence.

---

> ### Author Response · Authors · 2025-11-21
> **2. For Weakness 2 and Question 2**
>
> We appreciate the reviewer pointing this out. We agree that the decision to disable Hierarchical Merging (HM) for short videos is a critical design choice that **should be highlighted in the main text rather than placed in the appendix.**
>
> **2.1 Clarification on the Length-Adaptive Strategy**
>
> To address the reviewer's concern, we first clarify the rationale behind our length-adaptive configuration:
>
> - **For Long Videos**: The HM module is enabled. Enforcing hierarchical consistency (Event ⊃ Scene ⊃ Action) is essential here to **capture the complex hierarchical structures of long videos.**
>
> - **For short videos**:  Short videos typically have fewer events and a simpler temporal structure, meaning that preserving the diversity of independent features from multiple levels is more beneficial than imposing a rigid hierarchy. Disabling HM **preserves feature diversity** and yields better temporal localization.
>
> **2.2 Revisions in the Main Text (Sec. 3.3 & Sec. 4.1)**
>
> Following this rationale, we have moved the discussion from App. A.4.2 to the main text.

---

> ### Author Response · Authors · 2025-11-21
> **3. For Question 3**
>
> **3.1 Different roles: structure mining vs. query-aware aggregation**
>
> **- HTD merging (video-centric, offline).** HTD operates at the **video level** and is completely query-agnostic. Its goal is to mine the inherent temporal hierarchy from visual cues, producing physically and structurally coherent units (actions, scenes, events). This step significantly reduces the search space and regularizes the temporal structure, but by design it does not know where the semantic boundary of a specific text query lies. For example, a query like “the person cooks dinner” may span multiple HTD units (prepare ingredients, cook on the stove, serve the dish).
>
> **- Query-Centric CR Merging (Online).** This **semantic-driven** step utilizes a VLM to dynamically refine HTD candidates. Since a single semantic event (e.g., "cooking") often spans multiple structural units, relying solely on structural hierarchy is insufficient. CR Merging progressively re-aggregates adjacent HTD units based on **cross-modal similarity** and **intra-video consistency**, ensuring the final output captures the complete semantic scope of composite activities.
>
> In the revision, we briefly add this distinction and motivation to Sec. 3.5 and the appendix section “Distinction Between Merging Strategies” to make the roles of HTD (efficient, structure mining) and CR (query-aware semantic aggregation) explicit.
>
> **3.2 Why HTD merging alone is insufficient (and why CR is necessary)**
>
> CR is not a generic post-processing step, but a refinement module tailored to HTD segments.
>
> If we stop at HTD, each prediction is restricted to a single structural unit. This is efficient but too rigid: many natural-language queries correspond to composite activities that span multiple HTD units, so HTD alone often yields fragmented or truncated localizations. As shown in Tab. 4(revised paper), adding CR on top of HTD brings an improvement of 2.8% in mIoU on Tacos dataset.
>
> This two-stage design keeps VLM inference efficient (via HTD) while enabling accurate, query-aligned temporal boundaries (via CR).

---

### Author Response · Authors · 2025-12-03
**Comprehensive Rebuttal and Revision Summary**

We sincerely thank the reviewers for their constructive feedback. They commended HiTeA's groundbreaking results—**achieving state-of-the-art performance on long videos as a training-free** method while surpassing some supervised baselines. The reviewers also highlighted the novelty of the HTD module, which emulates human cognition to effectively bridge semantic understanding with temporal localization. Furthermore, they affirmed the method's **excellent generalization** across varying video lengths and the manuscript's clear exposition.

We detail our responses and revisions to the key points below.

**1.On Disabling Hierarchical Merging (HM) for Short Videos** *(Responding to Reviewer KVaa W2+Q2 and g3Bh Q3)*

Regarding Q3 from Reviewer g3Bh, we emphasize that this strategy is not a 'post-hoc' rationalization. Our work is primarily positioned for long-video, where the HM module is explicitly designed to capture deep temporal hierarchies. In contrast, short videos often exhibit a flatter structure that is prone to **Structural Degeneration** (where Event and Scene levels often collapse). Consequently, preserving the independence and diversity of features across levels is far more beneficial than enforcing rigid hierarchical consistency. Reviewer KVaa **affirmed this logic as a “very reasonable design choice.”** Following Reviewer KVaa’s suggestion to highlight this core design element, we have relocated the exposition of the 'Length-Adaptive Strategy' to the main text (Sec. 3.3 and Sec. 4.1).

**2.Rationale for the "Three-Layer" Hierarchy & Model Choices** *(Responding to Reviewer RwT5 Q1+Q5, g3Bh Q4, and wbqj Q1)*

We explicitly clarify the theoretical mapping between our 'Event-Scene-Action' hierarchy and the 'ViT-DINO-RAFT' backbones, corresponding to high-level semantic narrative, mid-level visual consistency, and low-level fine-grained motion, respectively.These models were selected as established representatives with proven generalization capabilities, a design choice explicitly validated by **new ablation studies in App. 4.3**. These results also confirm that our core contribution lies in the hierarchical processing strategy itself rather than in specific model choices. Furthermore, as mentioned in App. A.7, HiTeA does not enforce strict hierarchical nesting. The Candidate Refinement (CR) module is explicitly designed to handle this flexibility: it dynamically manages cross-level overlaps and merges based on specific queries, ensuring robustness in real-world domains.

**3.On Inference Efficiency & Runtime Analysis** *(Responding to Reviewer wbqj Q2, RwT5 Q6)*

Our feature extraction models operate offline and concurrently. While the proposed lightweight modules (HTD/CR) introduce negligible overhead ($\sim 10^{-3}s$), the primary latency stems from VLM inference. In the revised Sec. 4.5 and App. A.6, we include a new runtime analysis demonstrating that, thanks to this decoupled **offline-extraction/online-inference** design, the user-perceived online complexity is strictly $O(K)$ with constant $K$. On the long-video benchmark, HiTeA requires only 7–8s per query, significantly outperforming baselines (25s+). **Crucially, the inference latency remains stable and does not increase with video length.**

**4.New Experiments & Robustness** *(Responding to Reviewer RwT5 Q4, g3Bh Q2)*

We validate HiTeA's robustness on the challenging QVHighlights benchmark, **outperforming the previous best method**. We clarify that the system relies on a unified 'Two-Regime' configuration (Short vs. Long) rather than fragile per-dataset tuning, proving that the performance derives from principled architectural design rather than engineering burden.

**5.Other Clarifications** *(Responding to Reviewer KVaa W1+Q1+Q3, RwT5 Q2+Q3, and g3Bh Q1)*

(1) Per Reviewer KVaa’s suggestion, we clarified the "VLM-dominant" nature of $\lambda$. We also explicitly distinguishing HTD (offline, video-centric structural mining) from CR (online, query-aware semantic aggregation) to highlight their complementary roles in revised paper.

(2) We attribute the perceived instability on TACoS to the limited discriminative power of the **loose R1\@0.1 metric**, noting that the stricter mIoU metric confirms consistent performance. Additionally, we present decoupled experiments to refute the notion of 'component stacking', proving that HTD and CR are complementary modules that are both indispensable for the final performance.

(3) Addressing the works (UniTime-Zero, etc.) cited by Reviewer RwT5 and novelty concerns by Reviewer g3Bh, we explicitly define HiTeA’s position as **Strictly Fully Training-Free**,contrasting some methods still necessitate a training phase on external data.

Overall, we sincerely thank all reviewers for their constructive suggestions and the time dedicated to our work. Their insights have significantly strengthened our work. We also extend our gratitude to the Area Chair for their hard work in overseeing the review process.

---

### Meta-Review · Area_Chair_pghx · 2026-01-07

**Summary:**

## Summary
This paper proposes HiTeA, a training-free framework for long-video temporal grounding. Instead of relying on supervised training, HiTeA ultilizes pre-trained models and design a training-free framework to exploit these models for video temporal grounding.

## Discussion
During rebuttal, the author(s) address most concerns from the reviewers. AC agrees that the merits can outweigh the concerns, thus recommends to accept the paper.

**Reviewer Concerns:**

* Limited technical novelty (g3Bh)
* Method complexity and engineering burden (g3Bh)
* May not be able to handle non-hierarchical complexities (RwT5)

**Reviewer Scores:**

The paper receives the score of 8, 6, 4, 4 initially. After rebuttal, all reviewers keep their ratings unchanged.

---

### Decision · Program_Chairs · 2026-01-26

Accept (Poster)